# The Development of Immunotherapy for the Treatment of Recurrent Glioblastoma

**DOI:** 10.3390/cancers15174308

**Published:** 2023-08-29

**Authors:** Xudong Liu, Zihui Zhao, Wufei Dai, Kuo Liao, Qi Sun, Dongjiang Chen, Xingxin Pan, Lishuang Feng, Ying Ding, Shiyou Wei

**Affiliations:** 1College of Life Sciences, University of Chinese Academy of Sciences, Beijing 100049, China; liuxudongucas@outlook.com (X.L.); dssdo7@163.com (Y.D.); 2School of Medicine, Shanghai Jiao Tong University, Shanghai 200011, China; sjtuzzh123@sjtu.edu.cn; 3Department of Plastic and Reconstructive Surgery, Shanghai Key Laboratory of Tissue Engineering Research, Shanghai Ninth People’s Hospital, School of Medicine, Shanghai Jiao Tong University, Shanghai 200011, China; daiwfup@163.com; 4School of Biology and Biological Engineering, South China University of Technology, Guangzhou 510006, China; liaokuo@genomics.cn; 5School of Public Health, Li Ka Shing Faculty of Medicine, The University of Hong Kong, Hong Kong, China; q.sun@umcutrecht.nl (Q.S.); fenghku@connect.hku.hk (L.F.); 6Division of Neuro-Oncology, USC Keck Brain Tumor Center, University of Southern California Keck School of Medicine, Los Angeles, CA 90089, USA; dongjiang.chen@med.usc.edu; 7Department of Oncology, Livestrong Cancer Institutes, Dell Medical School, The University of Texas at Austin, Austin, TX 78712, USA; xingxin.pan@austin.utexas.edu; 8Department of Thoracic Surgery, West China Hospital, Sichuan University, Chengdu 610041, China

**Keywords:** recurrent glioblastoma, immunotherapy, CAR-T therapy, immune checkpoint inhibitor, cancer vaccine, oncolytic viral therapy

## Abstract

**Simple Summary:**

Glioblastoma (GBM) is the deadliest primary central nervous system (CNS) cancer in adults despite aggressive treatment. Once progressed, the prognosis is very poor and the effective traditional medicine treatment options are limited, so the management of recurrent glioblastoma (rGBM) remains challenging. Immunotherapy has revolutionized the prospects for many cancer types, but the intrinsic complexity of treating intracerebral tumors and the highly immunosuppressive environment have hampered the development of effective immunotherapies. The current focus of research in rGBM is on combination therapy, identifying predictive markers, and establishing synergy between immunotherapy and standard treatment. In this review, we discuss the current state of immunotherapy for rGBM, its future directions, and the challenges associated with each strategy.

**Abstract:**

Recurrent glioblastoma (rGBM) is a highly aggressive form of brain cancer that poses a significant challenge for treatment in neuro-oncology, and the survival status of patients after relapse usually means rapid deterioration, thus becoming the leading cause of death among patients. In recent years, immunotherapy has emerged as a promising strategy for the treatment of recurrent glioblastoma by stimulating the body’s immune system to recognize and attack cancer cells, which could be used in combination with other treatments such as surgery, radiation, and chemotherapy to improve outcomes for patients with recurrent glioblastoma. This therapy combines several key methods such as the use of monoclonal antibodies, chimeric antigen receptor T cell (CAR-T) therapy, checkpoint inhibitors, oncolytic viral therapy cancer vaccines, and combination strategies. In this review, we mainly document the latest immunotherapies for the treatment of glioblastoma and especially focus on rGBM.

## 1. Introduction

Gliomas have traditionally been classified by the World Health Organization (WHO) into grades I and II (low-grade gliomas) and III and IV (high-grade gliomas), based on their malignancy and histopathological features [1]. About 14.2% of all brain tumors are glioblastoma (WHO grade IV), the most common and most aggressive brain tumor [2]. Glioblastoma (GBM) is currently treated with surgical resection followed by radiotherapy and temozolomide (TMZ) [3,4]. Unfortunately, patients’ outcomes remain almost universally fatal with a median overall survival (OS) of 14.6 to 20.5 months, and older patients have a poorer prognosis, with an average survival of less than 8.5 months after diagnosis [5,6]. Since most patients experience recurrence, recurrent glioblastoma (rGBM) has a bleak outlook due to the lack of universally held treatment standards [7,8]. The Food and Drug Administration (FDA) approved only bevacizumab for the treatment of rGBM [9], but the randomized clinical trial (RCT) results with bevacizumab as first-line treatment for recurrence of GBM showed consistent results: A reduction in contrast enhancement intensity and volume, a reduction in peritumoral edema, a decrease in corticosteroid, and a statistically significant prolongation of progression-free survival (PFS), but no improvement in OS [10,11,12]. Due to the fact that 70~75% patients are not suitable for total resection at recurrence, resection has not been widely adopted or regarded as an effective treatment option [13,14,15]. Thus, new therapeutic strategies are urgently needed for rGBM.

In part, therapeutic failure results from extensive intratumoral heterogeneity of rGBM. The heterogeneity may be partially explained by the presence of GBM stem cells (GSCs), which are capable of self-renewal, differentiation, and plasticity [16]. These GSCs are generally resistant to chemotherapy and radiotherapy after primary GBM (pGBM) treatment, and seed the development of the therapy-resistant tumors [17,18,19]. Multiple studies have identified and isolated GSCs with tumor-initiating properties [20,21]. Tumor cells adjacent to GSCs will inhibit GSCs through paracrine and cell contact, so that GSCs enter a dormant state. Under certain circumstances, when non-functional GSCs are separated from surrounding tumor cells, their proliferative capacity will be reactivated leading to tumor recurrence [22,23]. Additionally, GBM has a poor prognosis due to the presence of the blood-brain barrier (BBB). There are two lines of defense for the BBB in the normal brain, a physical barrier and a chemical barrier, which can prevent macromolecular substances and unnecessary cells from entering the brain. As a result, GBM tumor cells are also unable to receive effective therapeutic drugs such as small molecules and antibodies. In addition, the central nervous system (CNS) has long been regarded as an immune-privileged site with restricting T cells to perform their functions [24]. This special microenvironment prevents T-cell priming and re-stimulation, thus impairs the anti-tumor immune response [25]. Moreover, GBM cells are able to induce local immunosuppressive effects through diverse mechanisms. For one thing, GBM cells can secrete many cytokines and/or chemokines to support the growth of tumor cells, affects the polarization of macrophage, facilitate the recruitment of regulatory T cell (Treg), and impede the maturation of dendritic cell (DC) and the function of natural killer (NK) cell. For another, GBM cells can also express molecules with immunosuppressive properties, such as programmed cell death protein 1 ligand (PD-L1), which hinders the proliferation and activation of T cell [26]. The immune surroundings with both pGBM and rGBM exhibit comparable suppressive alterations. A study verified that glioma-associated microglia/macrophages (GAMs), which are the principle immune cells infiltrating the tumor, showcase great inter- and intra-tumoral heterogeneity. These GAMs promote the accumulation of exhausted T cells, the infiltrating of Tregs, and the ineffectiveness of NK cells [27]. Hence, there are still opportunities and challenges to be met in finding more efficient treatments for rGBM.

Fortunately, decades of research into GBM’s molecular pathogenesis are now leading to innovative clinical trials. These trials benefit from advanced profiling of GBM’s genomic, epigenetic, transcriptomic, proteomic characterization, and interactions with the brain microenvironment and immune system [28]. Researchers have also achieved certain results in CNS drug delivery methods that have increased the survival of patients [29]. Recent advancement in immunotherapy, harnessing the body’s immune system to combat cancer, has led to significant clinical advances [30,31,32]. In 2013, the journal *Science* recognized cancer immunotherapy as the recipient of its prestigious ‘Breakthrough of the Year’ award, owing to the significant therapeutic advancements achieved through immune checkpoint blockade and chimeric antigen receptor-modified T (CAR-T) cells [30]. The 2018 Nobel Prize in Physiology or Medicine was bestowed upon the individuals who made the groundbreaking discovery of cancer therapy through the inhibition of negative immune regulation. This significant breakthrough has paved the way for the advancement of immunotherapy in clinical settings, leading to remarkable improvements in cancer patients’ outcomes. Notably, numerous immunotherapy drugs, such as monoclonal antibodies targeting cytotoxic-T-lymphocyte-associated protein 4 (CTLA-4), programmed cell death protein 1 (PD-1), and PD-L1, as well as CAR-T cell therapy, have received approval from the U.S. Food and Drug Administration (FDA) in recent years [30,31,33]. Despite the immune-privileged nature of the CNS, immunotherapy has made significant advancements in the treatment of GBM [34,35,36,37]. Ongoing research is exploring the potential of combination therapies [38,39]. Consequently, immunotherapy holds immense potential in the management of rGBM treatment. This review aims to provide an overview of current immunotherapy options for rGBM, encompassing vaccines, CAR-T, checkpoint inhibitors, and oncolytic virotherapy. Additionally, the challenges and future directions of these approach are discussed, offering a valuable resource for improving the OS of rGBM patients through immunotherapy.

## 2. Immunotherapy for the Treatment of Recurrent Glioblastoma

### 2.1. CAR-T Therapy

#### 2.1.1. The Background of CAR-T Therapy

T cells engineered to express chimeric antigen receptors to identify and attack specific markers expressed on the surface of malignant tumor cells have shown remarkable success in many tumors, particularly in hematological malignancies [40,41]. Tumor-specific CAR-T cells can be activated without antigen-presenting procedures and MHC molecules, which means they can be modified to accurately target most antigens in the human body [42,43]. In the last ten years, CAR-T therapies have transformed the management of many cancers with its high efficiency and fewer adverse events. Due to the protection of the BBB, the CNS used to be regarded as an immune-privileged environment in humans [44,45]. However, this strict mechanism was also found to be changed in some special situations, and then peripheral immune cells could enter the CNS from areas through high blood vessel regions like the choroid plexus and subarachnoid space. Particularly, when some pathogens invade or pathogenic damage happens, such as with some latent infections, the peripheral immune cells will cross the BBB and assist in keeping the homeostasis of CNS [46]. Previous studies have found the existence of various T cells in the tumor microenvironment (TME) of rGBM, while some tumor-related cells and factors in rGBM could inhibit the proliferation and protection function of these T cells [47,48]. Meanwhile, rGBM has some very special target substances, which are different from normal neurons and glial cells, making it suitable for CAR-T to identify and design. These characteristics of rGBM provide us with a theoretical possibility of CAR-T to effectively control the development of rGBM tumor cells with a slight neurotoxicity. Certainly, there are also many challenges in this process, just as the clinical practices tell. 

#### 2.1.2. The Latest Development in CAR-T Therapy

To date, there are many modified CAR-T therapies with different targets in vitro experiments, animal experiments, and clinical trials. The therapeutic targets of CAR-T that have completed clinical trials include EGFRvIII, IL13Ra2, HER2, etc., which are the most focused targets for rGBM. 

The first preclinical study of CAR-T therapy on GBM was conducted by Kahlon et al. in 2004 targeting the interleukin 13 receptor α2 (IL13Rα2) [49]. ILRα2 is highly expressed in GBM but has low expression in the normal brain and most normal tissues [50]. Therefore, IL13Rα2 became one of the most common targets for rGBM, and the first target treated in the clinical human body by CAR-T therapy. In 2015, Brown et al. conducted the first human trial in three rGBM patients targeting IL13Rα2 to explore the safety and effect of CAR-T therapy in rGBM [51]. This treatment was found to be well tolerated with a transient anti-tumor activity in two of the three patients. Although this phase 1 study finally failed to increase patients’ OS significantly, these findings provide the first promising human clinical experience for the treatment of rGBM with intracranial administration of IL13Rα2-directed CAR-T. The feasibility and safety of CAR-T for rGBM proved by Brown et al. successfully set the foundation for the future improvement of CAR-T therapy. Most recently, they also reported their phase 1 trial results for the off-the-shelf, allogeneic IL13Rα2-directed CAR-T product for the treatment of rGBM [52]. This allogeneic product was proven to have the feasibility, safety, and therapeutic potential for rGBM patients, which would dramatically reduce the costs of CAR-T therapies and increase their accessibility in the clinic.

Alongside IL13Rα2, EGFRvIII is another very interesting and well-known target for rGBM. EGRFvIII is a deletion mutation of the epidermal growth factor receptor (EGFR) and is often expressed in most tumors [53]. In GBM, about 40% of newly diagnosed patients have *EGFR* gene amplification, and about 50% of patients with EGFR-amplified GBM show constitutive activation of the oncogenic variant, EGFRvIII [54]. The first preclinical research for CAR-T targeting EGFRvIII in GBM was reported in 2009, which showed that the modified T cells have effective and specific cytotoxic activity against GBM tumor cells expressing EGRFvIII in vivo [55]. In the experiment by Donald et al. in 2017, clinical results of the EGFRvIII-directed CAR-T therapy were first observed. In their work, the EGFRvIII-directed CAR-T was intravenously injected into ten treated patients with rGBM [56]. All patients involved had a transient proliferation of CAR-T-EGFRvIII in peripheral blood. For seven patients who underwent further procedural intervention, it was found that CAR-T-EGFRvIII was successfully transported to the rGBM region, and antigen reduction occurred in five of these seven patients. Additionally, in Donald’s research they found no cross-reactivity of wild-type EGFR when patients used the CAR-T therapy. This further proved that CAR-T was a feasible and safe therapy for rGBM. And in May 2021, a successfully prolonged survival case following EGFRvIII-CAR-T treatment for rGBM was reported by Joseph et al. [57]. A 59-year-old patient, who was administered a peripheral infusion of CAR-T-EGFRvIII, exhibited a noteworthy survival of 36 months subsequent to GBM recurrence. This survival duration markedly surpassed the envisaged prognosis for rGBM. Moreover, the EGFRvIII-targeted T cells demonstrated sustained presence within her peripheral circulation for a span of 29 months during the ensuing follow-up period. This notable persistence duration stands as the most prolonged documented in the context of CAR-T trials for rGBM to date.

Another commonly studied target for rGBM is Human Epidermal Growth Factor Receptor 2 (HER2). HER2 is found to be overexpressed in many kinds of cancers and approximately 80% of GBM; however, it is also expressed to some extent in most normal tissues. The first preclinical research targeting HER2 by CAR-T therapy was published in 2010 by Ahmed et al. [58]. But until 2017, the results of the first clinical trials were firstly reported and showed that infusion of autologous HER2-directed CAR-T to 17 patients was well tolerated without dose-limiting toxicity in rGBM [59]. This clinical report also showed some clinical benefits of CAR-T therapy for rGBM patients involved through transient tumor reduction and/or tumor necrosis effects. Despite this encouraging result, considering the expression of HER2 in some important organs, the safety of HER2-directed drugs still needs to be the subject of more strict experiments in the future before it is widely used in clinic.

In addition to the above well-known targets, the B7-Homolog3 (B7-H3, also known as CD276), extracellular matrix metalloproteinase inducer (EMMPIRIN, also known as CD147), dissialoganglioside (GD2), matrix metalloprotease 2 (MMP2), CD133, CD70, etc., are other interesting targets in recent years for rGBM and have gradually entered different experimental stages [47]. For instance, in May 2022, a study reported that the use of CAR-T therapy targeting CD133 in mice with human GBM was considered successful because it reduced more than 80% of the tumor burden in these mice and successfully improved their survival rates [35]. Overall, although these different targets have performed well in vitro or animal experiments, there is still a distance for them to go in order to be truly clinically used and improve the OS of patients with rGBM. The main clinic research efforts into the effects of CAR-T on rGBM are listed below in Table 1, which contains the completed and ongoing trials but not the terminated ones.

#### 2.1.3. The Limitations of CAR-T Therapy

Despite the advantages and feasibility of CAR-T therapy, there are plenty of factors that hinder the application of CAR-T therapy in rGBM. 

(1) Whereas researchers have made numerous efforts in the molecular characteristics research into rGBM, only a few molecules remain suitable for further experiments. This is mainly due to the strong heterogeneity of rGBM. There is currently no target that can be ubiquitously present in all tumor cells and significantly distinguished from normal tissues. Therefore, we still need a more comprehensive and in-depth understanding of rGBM.

(2) The infiltration rate of T cells in rGBM remains inherently low due to the specificity of CNS and the protection of BBB. The peripheral immune cells thus are difficult to enter the CNS, including modified T cells by intravenous injection. Meanwhile, the TME of rGBM also has strong immunosuppressive effects on T cells, so learning how to enhance the chemotaxis and function of T cells is also one of the challenges [48]. 

(3) At present, the toxic and side effects of CAR-T are relatively small compared with other mainstream therapies, but some types of it still have dose toxicity when they are intravenously injected. For example, in a 2019 incremental dose experiment targeting EGFRvIII, there was a death case reported [60]. In addition, due to the specificity limitation of some targets, there are some problems such as off-target toxicity. For example, the common target HER2 is also expressed in some normal tissues of important organs. One of the main concerns of HER2-CAR-T therapy is the risk of attacking normal tissue [61]. Even so, the side effects of CAR-T treatment for rGBM are still acceptable. Most patients only suffer from transient discomfort. The systemic cytokine release syndrome (CRS), which is a common risk in CAR-T therapy, has not yet been reported. 

(4) In addition, CAR-T currently also has a certain drug resistance. Taking CAR-T therapy targeting EGFRvII as an example, although the continuous existence of CAR-T can be seen in peripheral blood and the tumor will be controlled in the short term, after the administration of EGFRvIII-specific CAR-T, the loss/down-regulation of tumor EGFRvIII occurs. New relapsed tumors thus lack the specific target, which will result in the failure of CAR-T [56]. Similarly, the same situation occurred after the administration of IL13Rα2-CAR-T [62]. In sum, although CAR-T therapy is currently being developed at various clinical stages, it has not yet resulted in a significant improvement and change of OS in patients with rGBM. However, we cannot judge the superiority of a treatment method merely by the OS increase. At the same level of OS, it would also be valuable if the CAR-T therapy could have less torturous adverse events or alleviate patients’ suffering compared with other treatments.

#### 2.1.4. The Prospectives of CAR-T Therapy

CAR-T therapy has great improvement possibilities in the future from the treatment methods and modes for rGBM. Optimization in multiple aspects could be carried out. 

(1) The development of better targets is needed to improve the specificity and efficiency of all targeted drugs including CAR-T. Given the high degree of heterogeneity in most solid tumors, a single effective target now is much insufficient. 

(2) Try to better recruit peripheral immune cells to GBM and increase the infiltration rate of CAR-T in the CNS. Some physical ways such as non-invasive micro bubble-enhanced focused ultrasound (MBF), or some biological ways like the up-expression of chemokines, are now proven in their ability to increase the permeability of BBB. And some chemotactic enhancement methods for CAR-T are also stable and feasible in preclinical experiments [63]. Moreover, to avoid consumption in the peripheral blood, direct local injection or intracavitary injection can be useful. And its safety and advantages have been shown in clinical experiments targeting multiple targets such as IL13Rα2 [64].

(3) In addition to the use of a single target, multiple rGBM targets can be used in combination. The combination of multiple tumor-specific targets, the combination of tumor-specific targets and anti-targets for normal tissues, the combination of tumor targets and targets for inhibitory cells, etc., can improve the targeting ability of CAR-T and reduce its off-target toxicity. For instance, Bryan and his colleagues formulated a bicistronic configuration aimed at facilitating the expression of a chimeric antigen receptor designed for EGFRvIII specificity, concomitant with a bispecific T-cell engager (BiTE) targeting the non-mutated EGFR variant. [65]. The treatment with this CAR-T secreting BiTEs effectively circumvented the phenomenon of antigen escape without observable toxicity and resulted in the nearly complete disappearance of GBM in mice. Niaz et al. developed a novel CAR-T targeting IL13Rα2 and EphA2 for enhanced GBM therapy and proved its tumor control effect better than any single targeting CAR-T [66]. 

(4) The combination of CAR-T therapy and other therapies can be adopted. For example, CAR-T is combined with chemotherapy or radiotherapy. Radiotherapy will release pro-inflammatory cytokines to increase the infiltration of immune cells into the tumor, and the radiation itself can also change the permeability of the BBB [67,68]. It has been found that a combination with radiotherapy can improve the efficacy of CAR-T therapy in rGBM and some other solid tumor models [69,70]. Moreover, the combination of CAR-T with some small molecule cancer inhibitors has also shown synergistic effects, such as with the tyrosine kinase inhibitor (TKI).

(5) Other ways to improve the function of CAR-T include the structure optimization of CAR-T or the control of the dysfunction effect in the rGBM TME. There are still many possibilities for CAR-T therapy for rGBM in the future. 

In conclusion, although current clinical CAR-T therapy, like many other treatments for rGBM, has not yet significantly increased the OS in patients with rGBM, its higher specificity, limited adverse events, and broad optimization space make it a new hope for rGBM. Still, there are many new-generation CAR-T therapy trials that have demonstrated high efficiency in the control of rGBM development in the preclinical phase and we believe that all of these efforts by humans to conquer cancer will eventually pay dividends in the future.

### 2.2. Immune Checkpoint Inhibitor

#### 2.2.1. CTLA-4 Inhibitors

In the tumor immune response, antigen-presenting cells activate T cells via MHC molecules and costimulatory signals, with the CD28/B7 pathway being an important costimulatory pathway. CTLA-4 (CD152) is expressed in both activated T cells and Tregs, and it functions as a potent competitive inhibitor of CD28/B7 as its affinity for B7 (CD80/86) is 10- to 20-fold higher than that of CD28 [71]. So generally, CTLA-4 acts as one of the immune checkpoints that inhibits T cell activation, and thus effectively inhibits anti-tumor immune response in TME. Studies have found that GBM patients with lower CTLA-4 expression on T lymphocytes tend to have a better prognosis [72], indicating CTLA-4’s value as a prognostic factor in GBM [73]. One of the earliest studies evaluating the effect of CTLA-4 blocker in glioma was conducted in 2007. The result showed that in glioma model mice, CTLA-4 blocking was linked to increased tumor-infiltrating T cells [74]. Subsequent studies in 2016 and 2019 demonstrated that CTLA-4 inhibitors, which disrupt the formation of the CTLA-4/CD80 complex within the tumor, improved the survival of GBM-bearing model mice [75,76]. Afterwards, several clinical trials have proven its efficacy and safety in tumor immunotherapy and more are ongoing [73,77,78,79,80].

Although pre-clinical trials have shown potential, and some antibody-mediated immune checkpoint blockades (ICB) of CTLA-4 have shown positive effects in patients with glioma [81], available clinical data on the use of CTLA-4 inhibitors as monotherapy in GBM have not been convincing to date. Currently, ipilimumab is the only CTLA-4 blocking antibody that has been approved by the FDA, but its efficacy in GBM has not been demonstrated yet. 

As for combination therapies involving CTLA-4 inhibitors, one clinical trial found that compared to using Nivolumab alone, a combination therapy of Nivolumab with ipilimumab showed lower tolerability and no obvious improvement of PFS/OS in patients with recurrent GBM [82]. In a case series study in 2016, 20 patients with rGBM were treated with ipilimumab and bevacizumab, and about 31% showed a partial response [83]. Some studies showed that ipilimumab may be particularly efficacious in patients with recurrent hypermutant GBM when applied in combination with other immunotherapy modalities in adjuvant and neoadjuvant settings [84]. Other clinical trials investigating the safety, tolerability, and efficacy of ipilimumab combination therapies in rGBM include NCT03233152, NCT04403649, NCT03707457, and NCT03430791. Another phase II trial (NCT02794883) evaluated tremelimumab (also an anti-CTLA-4 monoclonal antibody) and anti-PD-L1 antibody as monotherapies and combination therapies in patients with rGBM. About 41.7% of the patients treated with tremelimumab alone showed grade 5 disease progression, but only 18.2% with a combination strategy. The median OS for the tremelimumab group was 7.2 months.

Immunotherapy targeting CTLA-4 still faces some challenges, like adverse effects of CTLA-4 inhibitors and unsatisfactory therapeutic results in most GBM patients. CTLA-4 blockade monotherapy is not as effective in GBM as in other cancers owing to GBM’s unique characteristics. In the future, combination therapies could be a potential way out as T cells typically have multiple checkpoints. Moreover, further investigation of the CTLA-4 expression profile is needed to determine drug concentrations in clinical trials, and predictive biomarkers are also required to increase the efficacy of trials and therapies [85].

#### 2.2.2. PD1/PDL1

The PD-1/PD-L1 pair is one of the most representative ICBs that can reactivate T-cell function and promote anti-tumor activity upon inhibition. Due to encouraging outcomes observed in other malignancies [86,87], there is substantial interest in investigating the efficacy of PD-1/PD-L1 blockade in GBM including rGBM. In general, PD-1 blockade is mainly evaluated in CNS malignancies, including rGBM, as it does not necessitate crossing the BBB to locally inhibit the pathway.

Despite a study by Berghoff et al. indicating that PD-L1 is expressed in 72.2% of rGBM cases [88], PD-L1 inhibitor failed to meet the activity threshold when combine with VEGFR inhibitor axitinib [89]. In an open-label, randomized, multicenter phase III trial CheckMate-143 (NCT02017717), the use of PD-1 blockade as monotherapy did not demonstrate survival benefits compared to bevacizumab in patients with rGBM [90]. The median OS for the nivolumab group was 9.8 months while that of the bevacizumab group was 10 months. Similarly, in a phase II trial (NCT02337491), another agent, pembrolizumab, showed ineffectiveness as monotherapy or in combination with bevacizumab for treating rGBM [91]. However, a subgroup analysis revealed that patients with MGMT-methylated tumors and no baseline corticosteroid treatment had a median OS of 17 months, compared to 10.1 months for similar tumors treated with bevacizumab [90].

Some patients with hypermutated rGBM who have biallelic mismatch repair deficiency may benefit from PD-1 blockade [92], which is consistent with many other cancers [93,94]. However, Touat et al. suggested that the hypermutational burden induced by chemotherapy may not enhance the response to PD-1 blockade [95]. Additional trials are underway to further evaluate the responsiveness of hypermutated rGBM-NCT02658279, NCT04145115. While there is still controversy surrounding how to utilize mutation burdens to predict anti-PD-1 response, approaches that promote intratumorally lymphocyte infiltration are necessary for most patients.

Neoadjuvant administration of PD-1 blockade is one approach that has been proposed to enhance intratumorally lymphocyte infiltration in patients with rGBM. This approach may prime an effective systemic immunity, potentially facilitating local lymphocyte infiltration while the tumor is surgically removed [96]. Two trials have been conducted based on this hypothesis. One study (NCT02852655) with 35 rGBM patients found that neoadjuvant pembrolizumab led to a median OS of 13.9 months compared to 7.6 months for adjuvant pembrolizumab only [97]. In the other single-arm study (NCT02550249), 30 patients (27 rGBM and 3 ndGBM) were treated with nivolumab pre- and post-operatively, but the median OS for these patients was 7.3 months which was not superior to the existing strategy [98]. The Differences between the two studies could result from different drugs utilized, small numbers of participants, and/or selection bias while only certain patients with rGBM are eligible for additional surgeries. In a serial study, scientists noticed a population with enriched BRAF/PTPN11 mutations in 30% of rGBM cases that responded to PD-1 blockade [99]. Further investigation revealed that ERK1/2 activation in rGBM is favorable to PD-1 blockade and promotes tumor-infiltrating myeloid cells and microglia expressing more MHC class II and associated genes [100]. Another ongoing study (NCT02337686) is devoted to evaluating immune effector function in this neoadjuvant setting. Though extra caution is needed before drawing any conclusions, both studies demonstrated similar intratumorally and systemic immune changes, suggesting that combinations with other immune and non-immune agents may be worth exploring.

To date, the majority of studies have focused on combining anti-PD-1/PD-L1 with other treatment modalities. Some groups have focused on combinations with conventional methods like radiotherapy, chemotherapy, and anti-VEGF therapies, but with adjusted strategies. Novel procedures like stereotactic radiation (NCT04977375, NCT02866747, NCT02829931), laser interstitial thermotherapy (NCT02311582, NCT03277638, NCT03341806), and tumor treating fields (NCT03430791) are tested in multiple ongoing trials, with the hope of generating enough local immune reaction. Upregulation of multiple alternative immune checkpoints on T cells and/or tumor cells has been observed in other solid tumors as resistance to ICB [101]. Clinical trials targeting IDO1 including (NCT03532295), CTLA-4 (NCT02794883), LAG-3 (NCT03493932), and CD137 (NCT02658981), along with PD-1, are underway in rGBM. Combinations with other immunotherapies like tumor vaccine and oncolytic virus have generated plenty of interest with multiple ongoing trials (Table 2).

#### 2.2.3. Negative Immune Regulation

T cell exhaustion plays a significant role in the local immunosuppression and immune dysfunction observed in GBM. Worenieck et al. have unveiled T cell exhaustion signatures in various tumors and highlighted LAG-3 as one of the T cell immune checkpoints upregulated in GBM that lead to severe T cell exhaustion [102]. LAG-3 can inhibit the function of CD8^+^ T cells and enhance the immunosuppressive activity of Tregs [103]. Another study by Shen et al. showed that patients with LAG-3 expression on peripheral blood CD8^+^ cells exhibited poorer responses to ICB antibodies. Therefore, LAG-3 could serve as an independent biomarker to guide treatment as well as an actionable target for standard-ICB-resistant patients, significantly showing correlations with response, survival, and progression-free survival in various cancer types [104]. Clinical trials have already shown the anti-tumor activity of anti-LAG-3 agents, although modest [105]. Currently, a phase I clinical trial (NCT02658981) is evaluating the efficacy and safety of anti-LAG-3 agents in rGBM. Moreover, a previous study by Worenieck showed that compared to T cells with only one checkpoint, T cells expressing multiple immune checkpoints were more dysfunctional [102]. As a result, this phase I trial is also evaluating the combination of anti-LAG-3 and anti-PD-1, whose anti-tumor activity has already been demonstrated in a clinical trial involving unselected patients with cancer [106]. Initial data from another phase II/III trial (NCT03470922) has also shown that compared to using anti-PD-1 alone, melanoma patients receiving combination therapy showed improved PFS. In conclusion, LAG-3 is a unique, non-redundant checkpoint that limits the efficacy of standard ICB therapies such as PD-1 and CTLA-4 inhibitors. It holds potential as a biomarker that guides treatment, a candidate for novel agents or combinations, and a promising immune target for standard ICB-resistant patients.

T cell immunoglobulin and mucin-domain containing-3 (TIM-3) is another immune checkpoint involved in negative immune regulation. It was demonstrated to induce CD8^+^ T cell apoptosis and exhaustion, as well as inhibit T cell response in glioma [107]. This has led to disappointing outcomes in patients receiving anti-PD-1 therapy and a lower survival rate of GBM patients [108]. Currently, several phase I studies are underway to evaluate the potential of TIM-3 as a therapeutic target. One of these studies (NCT02817633) has reported tolerability and promising efficacy of TSR-022, an anti-TIM-3 monoclonal antibody, in patients with advanced solid tumors (AMBER). It may help patients who showed no response to standard ICB therapy [109,110].

Other immune checkpoints for negative immune regulation include T-cell immunoglobulin and ITIM (Immunoreceptor Tyrosine-based Inhibitory Motif) domain (TIGIT), VISTA, and B7-H3 (CD276) [111]. They are all potential ICB targets under research, but currently no trials are evaluating their efficacy in GBM. 

#### 2.2.4. Positive Immune Regulation

Inducible co-stimulator (ICOS) is a novel immune checkpoint and an independent prognostic factor for glioma. It is expressed on the surface of activated T cells and enhances the secretion of multiple immune cytokines [112]. ICOS participates in positive immune regulation as the ICOS/ICOSL pathway was shown to promote T cell differentiation, proliferation, and activation [112,113]. But on the other hand, it also induces Tregs activation, especially in GBM, in which its negative effects outweigh its positive effects [114,115]. ICOS thus played a negative role in the immune microenvironment of glioma and GBM through promoting tumor formation, development, and drug-resistance [116]. Wang et al. discovered a positive correlation between ICOS expression and glioma malignancy. In general, higher ICOS often indicates shorter life expectancy [115]. Therapeutic strategies targeting ICOS for glioma hold promise as it has already exhibited anti-tumor effects in some malignancies [117,118]. Wang’s work also revealed synergistic interactions between ICOS and other important immune checkpoints, suggesting the possibility for combination therapy. To date, several clinical trials have been testing a combination therapy of anti-ICOS and anti-CTLA-4/anti-PD/PD-1 [111,117,118]. Further studies and experiments are required to evaluate the efficacy and safety of anti-ICOS therapy in treating GBM.

Glucocorticoid-induced TNFR-related gene (GITR) and OX40 belong to the tumor necrosis factor (TNF) superfamily, and they also play positive roles in immune regulation. They reduce T cell apoptosis, boost T cell proliferation, and increase T cell activity. Until now, several agonist antibodies for GITR and OX40 are under investigation (NCT02598960, NCT02628574, NCT01862900) [111]. In general, many patients who receive ICB therapy targeting PD-1/PD-L1 and/or CTLA-4 have not shown promising responses thus far. But novel immune checkpoints listed above show promise in improving the situation. They may offer potential benefits for patients who have exhibited unsatisfactory responses to standard therapy. They also have their own advantages over standard ICB. For example, the intracellular tail of TIM-3 has no ITIM or immunoreceptor tyrosine-based switch motifs (ITSM) [108]. Moreover, many of them are directly implicated in the progression of GBM and are involved in immune response recruitment and activation. Although currently there are not many studies investigating their therapeutic efficacy in GBM, hopefully novel immune checkpoints may be of greater importance and become the focus of future research. 

#### 2.2.5. Challenges and Future Directions of ICB in rGBM

Unlike other immunotherapies, ICB is extremely dependent on the intact immune system, from antigen presentation to effector lymphocytes activation. This is the major challenge in achieving positive results in rGBM as monotherapy given the local and systemically suppressed immune environment created by GBM [119]. Specifically, T cell dysfunction has been considered a hallmark of GBM, which would not be an easy fix by ICB [48]. Furthermore, immunosuppressive therapies such as chemotherapy or steroids that rGBM patients may go through could further limit the benefits of ICB [120]. Additional constraints unique to CNS tumors include the restricted access of drugs to the CNS. Many trials have attempted to circumvent this issue by applying ICB directly within the tumor. Duerinck et al. tested the idea by injecting ipilimumab and nivolumab intracerebrally in 27 patients (NCT03233152) [121]. The treatment appears to be safe and feasible, with a median OS of 9.5 months. Further studies are needed to determine whether local administration within tumors is required for optimal efficacy. 

A simple modification to treatment regimens may help the situation, as neoadjuvant treatment appears to be an attractive strategy for rGBM. Despite the lack of responses or partial responses in OS, pro-inflammatory changes in the tumor microenvironment are encouraging. It is possible that other checkpoints may be more predominant in rGBM, and thus PD-1 blockades may only improve lymphocyte activation without reversing the effects controlled by other checkpoints. Thus, novel checkpoints such as VISTA [122], Siglec-15 [123], and HHLA2 [124] may be worth testing once their role in rGBM is confirmed. Nevertheless, ICB seems to be a promising addition for many current immunotherapies relying on cytotoxic T cell functions with highly expressed intertumoral immune checkpoints in rGBM. In turn, other immunotherapies may compensate for the limitations of ICB by presenting antigens, creating a local immune response, or overcoming the immunosuppressive tumor microenvironment. The search for biomarkers to identify patients who are more responsive to ICB is also a promising avenue for further exploration.

### 2.3. Cancer Vaccination Therapy for rGBM

#### 2.3.1. The Background of Cancer Vaccination Therapy for rGBM

The use of anti-tumor vaccines, another form of immunotherapy, has also garnered significant interest in the treatment of rGBM due to its demonstrated potential and promise in both preventive and therapeutic effects [4,125]. This therapeutic approach typically targets tumor antigens to elicit adaptive immune responses against cancers. Due to the relatively low tumor mutational burden (TMB) observed in rGBM, the antigen targets selected are typically tumor-associated antigens, with only a minority of mutations serving as tumor-specific antigens (TSA) [81]. According to the immune subtypes of GBM classified by Han Lin et al., immune subtype 3 (IS3) exhibits the poorest prognosis but derives the greatest benefit from vaccination therapy [126]. Overall, numerous vaccination approaches are currently under investigation [127], with the majority still in the early stages of clinical development and clinical trials.

Generally, GBM vaccines are classified into several groups, including peptide vaccines, immune cell-based vaccines (DC cell-based, B cell-based), and nucleic acid vaccines. Table 3 presents the primary vaccines that have been studied or tested in rGBM. Next, we will provide a detailed discussion of each type of them.

#### 2.3.2. Peptide Vaccines

Peptide vaccines typically consist of peptide spanning 8–30 amino acids. These vaccines function by encompassing TSA or TAA (tumor-associated antigens).

Among the TSA peptide vaccines, Rindopepimut (CDX-110) has garnered significant interest, which is characterized by low off-target toxicity. However, its patient eligibility is limited as it specifically targets EGFRvIII, a mutated variant of EGFR found only in 25–30% of GBM, with 82% of tumors not expressing it upon recurrence. Several clinical trials have been conducted to evaluate its efficacy. 

Early studies include three uncontrolled phase II trials reported in 2010, 2011, and 2015. In these trials, GBM patients who had undergone complete surgical removal and chemoradiotherapy were administered rindopepimut vaccination. The results showed a median OS of 24 months, representing a modest improvement over historical control [128,129,130]. In a phase II trial in 2015 by Reardon et al., the combination of rindopepimut and bevacizumab was shown to have promising therapeutic activity and tolerability in patients with rGBM [131]. In 2017, Weller M et al. reported that patients with minimal residual disease who received rindopepimut with TMZ did not show an improvement in OS compared with patients receiving TMZ alone in a double-blind, placebo-controlled, multicenter phase III trial ACTIV, Nonetheless, the result did exhibit notable humoral immune response [132]. In 2020, Reardon DA et al. reported favorable outcomes when exploring the effectiveness of rindopepimut in combination with bevacizumab in a smaller cohort of EGFRvlll-positive GBM cases [133]. Collectively, these investigations imply that rindopepimut could potentially exhibit certain levels of efficacy in meticulously chosen patient subgroups. However, additional research is imperative to ascertain the optimal treatment regimen. Those contradictory and inconsistent results questioned the effect of the single antigen-targeted vaccine and lend support to combination strategies and multi-epitope vaccines.

Isocitrate dehydrogenase-1 (IDH1) mutations, on the other hand, create TSA as a potential target for vaccination therapy. The frequency of IDH mutations was found to be less than 10% in pGBM, whereas it exceeds 70% in rGBM, thus indicating a broader application regimen for rGBM compared to vaccinations targeting EGFRvIII. Preclinical studies have already shown that peptide vaccines spanning the IDH1 mutation can provoke antitumor T cell reactions. A phase I clinical trial reported that around 90% of glioma patients exhibited an immune response subsequent to therapy with a vaccine targeting IDH1- R132H^+^ in 2021 [134]. Combination therapy involving PD-L1 checkpoint inhibition has also been proposed [135].

Wilms’ tumor 1 is another notable antigen in GBM, with a particularly high presence reaching 94% [136]. Unlike other single antigen-targeted vaccines, the risk of immune escape is relatively low for a WT1 vaccine, as the loss of WT1 expression was shown to halt tumor proliferation and induce cancer cell death. In 2020, J.D. Rudnick et al. reported the clinical responses to WT1 vaccination among patients with rGBM who were positive for human leukocyte antigen HLA-A24 in a phase I/II study. The results were limited with a 9.5% overall response rate and 20 weeks of PFS time [137].

The vaccination approaches discussed above are all single antigen-targeted, but multiple-epitope peptide vaccines are believed to hold greater potency and efficiency due to their ability to induce more robust and comprehensive immune responses.

IMA950 represents an innovative peptide vaccine comprising 11 synthetic tumor-associated antigens. This formulation facilitates the activation of specific cytotoxic T lymphocytes (CTLs) aimed at eliminating malignant tumor cells. A phase I/II trial completed in 2019 evaluated IMA950 in combination with poly-ICLC and TMZ. The PFS of patients in the overall cohort were 93% and 56% at 6 and 9 months, respectively [138]. However, IMA950 has not shown any benefit in rGBM patients so far [127]. It is worth mentioning that the array of peptide set selected from the IMA950 may have potential applications in the immunotherapy of high-grade gliomas, which is different from other peptide vaccines [139].

TAS0313 is a multi-epitope long peptide vaccine targeting multiple TAAs in rGBM. In 2022, it was demonstrated to have encouraging effectiveness and favorable safety profile among rGBM patients [140].

Heat-shock protein peptide complex-96 (HSPPC-96) is another vaccine approach that targets multiple tumor antigens. In 2014, Bloch et al. delved into the efficacy and safety of HSPPC-96 vaccination within in a phase II trial involving rGBM patients and reported a median OS time of 42.6 weeks [141]. Another phase I trial showed a 2.3-folds increase in the tumor-specific immune response of ndGBM patients after they were treated with the HSPPC-96 vaccine [142]. Presently, numerous researchers are investigating the possibility of HSPPC-96 in combination with radiotherapy and chemotherapy (NCT00905060) in treating primary GBM, as well as in conjunction with bevacizumab (NCT01814813) for rGBM patients [143].

The limited expression of GBM-specific antigen due to low TMB and extensive heterogeneity of GBM between individual patients has been posing challenges for GBM therapy, which means there is not a one-fit-all vaccination approach. Recent strides in innovative bioinformatics tools and next generation sequencing, however, enable us to systemically discover tumor-specific neoantigens as suitable targets, which have the potential to solve the problem and are thus garnering significant attention. Through whole exome sequencing of patient tumor cells and peripheral blood, we can explore expressed mutations in tumors and then rank candidate targets for synthesizing to generate vaccines [144]. Those personalized, neoantigen-based vaccines have shown robust tumor-specific immunogenicity along with initial indications of anti-tumor activity in patients with melanoma and various other cancer types [145]. Moreover, it elicits much lower toxicity compared to TAA-targeted vaccines. Building upon these encouraging outcomes, two phase I/Ib studies of multi-epitope, personalized antigen vaccines were carried out and reported in 2019, in which Keskin et al. highlighted the generation of circulating polyfunctional neoantigen-specific CD4^+^, CD8^+^ T cells that were enriched in a memory phenotype and found an increase in tumor-infiltrating T cells (TILs). It suggests that neoantigen vaccines have the potential to remodel a “cold” tumor environment into an immunologically active “hot” one. The study also provided evidence that neoantigen-specific T cells can migrate into an intracranial GBM. But disappointingly, all patients in the trial still experienced tumor recurrence and ultimately died [146]. In another similar phase I trial conducted by Hilf et al., comparable findings were reported, showing acceptable safety profile and sustained T cell response [147]. In 2019, a phase III trial on HLA-A24 positive rGBM patients investigating personalized peptide vaccination was conducted, but neither the primary endpoint (OS) nor the secondary endpoint was achieved [148]. Other trials have also investigated the safety of combination therapy with radiation therapy or immune checkpoint inhibitors (ICIs) [144,149]. In conclusion, this strategy requires further exploration of its efficacy and more improvement to overcome challenges like tumor-intrinsic defects and immunosuppressive factors in the microenvironment. Combination therapies may offer a potential solution to address these obstacles. It is also noteworthy that the process of neoantigen identification and vaccination development is time-consuming (about 3 months) [150], which poses another limitation to its application. Detecting recurrent and shared neoantigens holds promise in addressing this issue. Subunit vaccines exhibit commendable safety profiles and noteworthy efficacy, rendering them viable options [151]. In comparison to whole protein or pathogen vaccines, these domain-based vaccines offer notable advantages. In 2021, Mahmoud Gharbavi et al. reported that they designed and synthesized a multi-domain recombinant vaccine forGBM. The process involved the selection of the most potent domains of TAAs using immune-informatics analysis and their combination to elicit an immune response in the host, whereby the potency of this novel multi-domain subunit vaccine was demonstrated through physicochemical analysis, and its antigenicity was estimated at 0.78. This multi-domain vaccine holds the potential to offer both preventive and therapeutic advantages [152].

The Mannan-BAM, TLR Ligands, Anti-CD40 Antibody (MBTA) vaccine represents another personalized vaccination approach that targets multiple TSA. This vaccine offers distinct advantages because it circumvents the long process of silico tumor-neoantigen enrichment required for personalized neoantigen peptide vaccination by enabling the in vivo processing of tumor-specific neoantigen via endogenous pathways, thus triggering activation of the innate immune system [153]. In this way, it allows the innate immune system to naturally identify antigenic targets through via inherent processing mechanisms. Furthermore, the MBTA vaccine has shown potential to overcome the challenges associated with immunosuppression and intratumoral heterogeneity [154].

#### 2.3.3. Cell-Based Vaccines

Currently, about half of the ongoing phase II/III trials on GBM involve cell-based vaccines, the majority of which use a DC carrier. Other cell-based vaccines include B cell-based vaccines that have also gained much attention due to their high mobility and convenience to be manufactured ex vivo [136]. The subsequent section provides comprehensive descriptions of these vaccine types.

##### Dendritic Cell (DC) Vaccines

DC vaccines are of great interest due to the critical role played by DCs in immune regulation and antigen presentation. They can target tumor antigens directly or serve as immune-boosting adjuvants in vaccination therapy [155]. DC vaccination directed towards tumor peptides has exhibited promising outcomes in the treatment of rGBM patients Adjuvant DC immunotherapy in rGBM patients was also shown to induce long-term survival. Typically, DC vaccination was generated ex vivo from DCs harvested from patients and subsequently stimulated by either tumor antigens, cell lysates, recombinant proteins, or nucleic acids before administration. The commonly utilized DC types include Mo-DCs and leukemia-derived DCs (DCleu).

Several studies have revealed the clinical efficacy of DC-based vaccines, but there have also been conflicting results. In 2020, a phase II clinical trial of alpha-type-1 polarized DC-based vaccination have showed a significant survival-prolonging effect in newly diagnosed high-grade glioma patients [139]. Vaccination using DCs loaded with TAAs and mRNA of neoantigens extended patients’ mOS to 19 months [156]. In 2022, a meta-analysis encompassing 15 clinical trials (comprising 452 cases and 629 controls) was conducted to evaluate the efficiency of DCV in newly diagnosed GBM (ndGBM) patients, which revealed that DCV had no impact on 6-month PFS or 6-month OS but led to significantly longer 1-year OS and longer 2-year OS. Its delayed effect suggests the necessity for additional therapies to facilitate the earlier action of DCV [157].

However, two meta-analyses in 2021 concluded that DCV has no obvious impact on the prognosis of ndGBM. But those two analyses had relatively small sample sizes, which may have influenced the conclusions drawn [4,158].

Furthermore, the observed heterogeneity in the results of DCV studies may also be attributed to variations in methods employed and differences in patient populations recruited. Studies have indicated that patients with low B7-H4 expression who received DCV treatment experienced significantly prolonged OS. Furthermore, methylated MGMT promoter, wild-type IDH, and mutation-type TERT are also linked to better response to DCV [159]. The relatively short life expectancy for GBM may obscure the impact of DCV too as it typically requires a minimum period of 6 months to become evident. Based on these studies, stratification of GBM patients based on molecular biomarkers to identify more sensitive groups may be necessary prior to DCV therapy.

Among the single targeted DC vaccine candidates, Wilms’ tumor 1 (WT1)-pulsed autologous DCs and cytomegalovirus phosphoprotein 65 RNA (CMV pp65)-pulsed DCs have shown promise. The efficacy and safety of the WT1 DC vaccine in rGBM patients were already demonstrated in a phase I trial [155]. Researchers have also found that compared to WT1 peptide vaccination therapy, DC-based vaccination induces and activates more tumor antigen-specific cytotoxic T cells in rGBM, which may lead to prolonged survival in rGBM patients. Several phase I trials of CMV pp65 DC vaccine have shown promising results as well [160], and currently a randomized phase II trial is recruiting newly diagnosed GBM patients (NCT02465268).

However, as rGBM is highly heterogenous, several studies have revealed that vaccines targeting a single tumor antigen encounter challenges in attaining optimal clinical outcomes unless the antigen is extensively expressed in tumor cells. Therefore, there is a growing focus on the development of vaccines that target multiple antigens.

ICT107 is a DC vaccine pulsed with six synthetic peptides. It is specifically designed for GBM and is produced through the ex vivo incubation of patient-derived DCs with six GBM TAAs. Its safety and therapeutic potential in patients possessing the HLA-A2 marker have already been demonstrated in some early phase clinical trials, which led to a phase III trial carried out in HLA-A2 positive patients with ndGBM (NCT02546102). However, this phase III trial was suspended in 2017 due to inadequate funding, halting further progress in its evaluation.

The autologous tumor cell lysate-pulsed DC vaccine can target multiple antigens too. This personalized vaccination therapy also addresses the heterogeneity of GBM by utilizing patient-derived autologous antigens rather than standardized antigens. DCVax-L, for instance, employs autologous whole tumor lysate to pulse patient-derived DCs, targeting the full repertoire of antigens and minimizing immune escape. Theoretically, this kind of vaccine should be more efficient but carry a higher risk of autoimmune response. As promising results have been observed in preclinical models and early-stage clinical trials, a phase III prospective externally controlled cohort trial (NCT00045968) was conducted in ndGBM. By 2018, this phase III trial showed that the overall intent-to-treat (ITT) population exhibited a median OS of 23.1 months from surgery and a low incidence of grade 3 or 4 adverse events (2.1%), superior to the median OS of 15–17 months reported in past studies and clinical trials [161]. In 2023, the same trial reported that the median OS for ndGBM patients treated with DCVax-L stood at 19.3 months, whereas the control group exhibited a median OS of 16.5 months. The 48-month survival rate from randomization was 15.7% compared to 9.9%. For rGBM patients, DCVax-L also showed advantages compared to the control group. Moreover, a better response was observed in patients with methylated MGMT [162]. This study demonstrated that the incorporation of DCVax-L alongside SOC led to a clinically meaningful and statistically significant prolongation of survival in both ndGBM and rGBM patients, which were notably superior when contrasted with external controls that only underwent SOC. Overall, the addition of DCVax-L to standard therapy has shown feasibility, safety, and the potential to extend survival in GBM patients. Another randomized phase II trial (NCCT03014804) on Vax-L is currently in progress.

AV-GBM-1 is an autologous tumor-initiating cell pulsed DC vaccine, which is different from DCVax-L (utilize fresh whole tumors). A multicenter phase II trial was designed to evaluate AV-GBM-1 and reported that the treatment was well-tolerated with a prolonged median PFS, though no median OS improvement was observed [163]. Another phase III trial for AV-GBM-1 has been approved by the FDA and is underway (NCT05100641).

Similar to the advantages of neoantigen-targeted peptide vaccines over TAA peptide vaccines, personalized neoantigen-pulsed DC vaccines have also been considered more effective than TAA-pulsed DC vaccines [164]. Numerous trials utilizing personalized neoantigen-pulsed DC vaccines are currently ongoing [165].

Combinatorial therapy of DC vaccines with chemotherapy and checkpoint inhibitors is also under active research, as it has been demonstrated that the efficacy of DC vaccines enhanced through this approach [166].

Although the administration of inactivated tumor cells or patient-derived tumor cell lysates have exhibited superiority, their efficacy is hampered by their inability to kill tumor cells before inducing immune responses, which can be fatal as GBM progresses rapidly. In 2023, Chen et al. developed a bifunctional cancer cell-based vaccine (therapeutic tumor cells) that facilitates direct tumor killing and antitumor immunity simultaneously. It represents a promising cell-based immunotherapy as it has shown therapeutic efficacy in a recurrent GBM mice model [167].

Finally, DC vaccine immunotherapy still faces several challenges, including the presence of an immunosuppressive TME, plus intrinsic drawbacks like high costs as well as time-consuming processes, which limit its widespread application [168]. However, despite all of these challenges, DCV still represents a promising new strategy for GBM and other malignancies with validated safety and feasibility.

##### B Cell Vaccines

B cell vaccines are another type of emerging cell-based vaccine for GBM that harbor great potential. Lee-Chang et al. developed BVax, which was shown to migrate into secondary lymphoid organs to activate T cells for the removal of GBM cells. In a trial conducted on GBM model mice, the combination of PD-L1, BVax, and radiation therapy led to 80% tumor eradication and sustained potent immunological memory, effectively preventing tumor re-growth [136].

#### 2.3.4. Nucleic Acid Vaccines

Nucleic acid vaccines, including mRNA vaccines and DNA vaccines, offer several advantages over peptide vaccines. For instance, they can encode entire tumor antigens and are not constrained by the patient’s HLA type compared to conventional vaccination [169]. Additionally, they have the capability to deliver multiple antigens and exhibit greater resistance to drug resistance [170]. Moreover, the production of nucleic acid vaccines could be more rapid and cost-effective when compared to peptide vaccines.

In 2022, S. Amit et al. employed the UNITE platform to develop a multi-antigen targeted DNA vaccine (ITI-1001) encoding human cytomegalovirus (HCMV) proteins that are expressed in GBM cells. The vaccine elicited robust humoral and cellular immune responses and led to improved survival in GBM-bearing mice [171]. This therapy is particularly suitable for certain patients whose medical conditions do not allow leukapheresis and autologous DC immunity. In the same year, a combination therapy involving the DNA vaccine pTOP and immune checkpoint blockades in orthotopic unresectable GBM model mice was shown to improve effector T/Treg ratios and infiltration of CD8 T cells in tumor, opening a new prospective for GBM treatment [172].

Compared to DNA vaccines discussed above, mRNA vaccines have higher expression efficacy and are easier to design and modify, making them well-suited for individualized treatment approaches. Moreover, mRNA vaccines offer enhanced safety as they do not require integration into the patient’s genome. The efficacy of mRNA vaccines has been evaluated in various types of tumors, yielding promising results. In 2022, Han Lin et al. reported using gene expression profiling interactive analysis (GEPIA) to evaluate the expression profile of GBM antigens as well as their clinical influence. They selected six TAA and TSA that were highly correlated with GBM prognosis to be potential targets for developing mRNA vaccines and found that GBMs of the immune-cold subtype I3 were more likely to benefit from vaccination. Thus, screening mRNA-sensitive patients (for example, IS3) before treatment is important [126]. Also in 2022, another similar research effort selected nine antigen candidates, adding to the previous research [173]. 

#### 2.3.5. Limitations and Strategies to Enhance Cancer Vaccines for rGBM

As for vaccination therapy for rGBM, there are still many challenges waiting to be addressed, including: (I) systemic and local immunosuppression in the tumor microenvironment, (II) high tumor heterogeneity and deficiency of specific tumor antigens (due to low TMB) within GBM [174], (III) BBB which prevents peripheral immune cells from entering CNS, (IIII) severe adverse effects of some vaccines. Efforts have been made to overcome these challenges and we compile some possible ways below. For example, to overcome the local immunosuppressive environment, studies have demonstrated that certain agonists targeting tumor-associated macrophages (TAM), such as poly-ICLC, resquimod, and imiquimod, can be used as vaccine adjuvants to enhance the efficacy of vaccine therapy. It can prolong the median PFS of GBM patients to 21 months post-diagnosis [175]. The underlying mechanism is that these agonists can repolarize TAM, which makes up 80% of immune cells in the tumor microenvironment. M2 phenotype TAMs, in particular, contributes to tumor progression and invasion through several mechanisms [176]. Another strategy to make TME “hotter” is to utilize personalized neoantigen-targeted vaccination therapy [146]. Furthermore, accumulating evidence shows that the gut microbiota can regulate immunity and metabolism within the GBM microenvironment thus making it a potential therapeutic target to modulate the immunosuppressive TME of GBM too [143]. To find TSA and overcome intertumoral heterogeneity, personalized neoantigen-targeted vaccines hold promise with effectively reduced off-target toxicity [165]. To avoid immune escape and solve the problem of individual heterogeneity, we may utilize tumor cell-pulsed DCV or add other therapeutic modalities like molecular targeted therapy to immunotherapy. To disrupt BBB and enable the access of immune cells, combination therapy with MRI-guided laser ablation (MLA) may be beneficial [177]. However, despite these promising results observed in preclinical investigations and early-phase clinical trials, as well as instances of success in isolated case reports, the transition to phase II/III trials is still notably demanding. To date, there are no successful phase III clinical trials with large patient cohorts for immunotherapy of GBM [38].

In conclusion, vaccination therapy has been considered one of the most promising avenues for enhancing the outcomes of rGBM patients. From trials that have been conducted so far, it is evident that single-agent immunotherapy has limited efficacy for rGBM, so rational combinatorial treatment strategies are worth more attention. In the future, it is imperative to further deepen our comprehension of the mechanisms underlying immunosuppression in GBM Additionally, there is a pressing need to develop more potent and efficacious tumor-specific antigenic profiles. Finally, although several vaccines have already shown efficacy and safety in phase I and II trials, the overall results of phase III clinical trials are still disappointing, without significant improvement in the prognosis of rGBM. Accordingly, more phase III trials are needed.

### 2.4. Oncolytic Viral Therapy in Recurrent GBM (rGBM)

In recent years, oncolytic virus (OV) therapy has demonstrated great potential in prolonging survival, improving patients’ quality of life, and lessening adverse effects. In contrast to OV, following traditional therapy, such as surgery, radiotherapy, or chemotherapy, the median survival of patients suffering from pGBM is approximately 14.6 months [125].

The clinical trials and animal experiments evidence is shown in Table 4.

The mechanism of the oncolytic virus is still unclear and the oncolytic procedure is multi-related and multi-staged. Nevertheless, there are two dominant perspectives: one is that OVs directly destroy GB cells, and the other is that OVs induce tumor cell lysis by virus-specific infection of tumor cells and the release of viral progeny to induce tumor cell lysis [4,126,127,134,137,162,173].

There are four prominent OV families tested in human or animal trials, which are Herpes simplex virus-1 based (HSV-1-based), AdenovirusBased, ReovirusBased, and Newcastle Disease VirusBased.

#### 2.4.1. Herpes Simplex Virus-1 Based (HSV-1-Based)

HSV-1 is a large double-stranded DNA virus, a common human pathogen with a long-term latent and lifelong potential for infection in humans [146]. It is a neurotropic virus, and the genes involved in tumor lysis differ from neurotoxic genes, allowing tumor cells to replicate and manipulate tumor lysis genes [147] conditionally.

Currently, three HSV-1 lysosomal strains (including HSV1716, G207, and G47Δ) have completed phase I clinical trials in glioma patients and clinical trials to evaluate the efficacy and safety of two other HSV-1 lysosomal strains (M032 and QNestin34.5) are ongoing [149].

##### HSV1716

HSV1716 is a double-copy neurotoxic gene γ134.5-deficient generation lysogenic HSV that selectively replicates in actively dividing cells [150]. In 2000, R. Rampling et al. reported the first evidence in support of the safety of HSV1716 for rGBM treatment in humans [148]. In that study involving nine patients who had previous surgery and radiotherapy, three each received 10^3^, 10^4^, and 10^5^ pfu of HSV 1716 by stereotactic injection directly into the tumor. Five of nine died after the injection from 8 weeks to 9 months during the follow-up. Three underwent further surgery; one died of tumor progression at nine months, and two were alive and well at 17 months. The other two patients remained well at 14 and 24 months, respectively. They concluded that it is feasible to use replication-competent HSV as part of an rGBM combination regimen.

##### G207

G207 is a double-copy γ134.5 gene deletion and insertion of the exogenous gene lacZ into the UL39 gene [131], thus inactivating ICP6 which supports conditional replication of the virus in actively dividing cells. The effectiveness was demonstrated in mouse and non-human primate experiments [145,151,152,153]. James M. Markert et al. showed the safety of inoculating G207 in the brain surrounding a glioma resection cavity [128]. The maximum dose in this 1b trial (registration number: F05041106) is 1 × 10^9^ pfu. Three of the six subjects improved Karnofsky’s performance following the G207 injection. The median survival was 6.6 months (range: 2–20.75 months) from G207 inoculation. No patients did further chemotherapy, which indicated G207 administration in any decrease in tumor progression. None of the deaths or complications could be attributed to G207 administration in the tumor or brain tissue next to the resection cavity. Five years later, this research group conducted a phase 1 trial (registration number: NCT00157703) to show the safety and potential clinical response of single-dose stereotactic intratumoral administration by G207 in rGBM patients [129]. Nine patients received one dose of G207 and then were treated focally with 5 Gy radiation. Six patients had stable conditions or partial response for at least one point. The PFS was approximately 2.5 months (95%CI: 1–5.75), and the estimated median survival time was 7.5 months (95%CI: 3.0–12.7) from G207 injection. One year later, an American team reported that a 52-year-old Caucasian female extended a tumor progression-free interval of 6 years with G207 oncolytic therapy and brief exposure to further treatments after the first treatment doing aggressive tumor resection, radiotherapy, and chemotherapy [130]. Recent gene research has revealed that the immune activity differences in post-G207 and pre-G207 samples are associated with survival duration in patients with rGBM. The tremendous change following the G207 injection is the increasing proportion of CD4, CD8, and CD8^+^ T-cell to exhausted CD8^+^ T-cell ratio, and the NK CD56 dim to total tumor-infiltrating lymphocytes ratio. The survival data showed that four of six survived longer than the median survival of GBM recurrence, four months.

##### G47Δ

G47Δ was constructed by deleting the α47 gene in G207 viral mutant [154]. Tomoki Todo et al. have published their newest results of a phase 2 trial (registration number: UMIN000015995) for applying G47Δ in residual or rGBM treatment in 2022 [132]. The research showed the median OS was 20.2 (95%CI: 16.8–23.6) months after G47∆ initiation and 28.8 (95%CI: 20.1–37.5) months after the primary surgery. 17 of 19 patients suffered from fever as the most common adverse event. The only serious side effects (grade 2) occurred in one patient (5.3%), leading to a prolongation of hospitalization. G47Δ therapy indicated good efficacy and safety in rGBM treatment, which approved it as the first oncolytic virus product from the Japanese Pharmaceuticals and Medical Devices Agency.

##### Genetically Engineered Herpes Simplex Virus Expressing Interleukin-12 (M002)

James M. Markert et al. compared M002 with R3659, R8306, and G207 and found that M002 indicated superior antitumor activity, with no significant imaging or clinical evidence of toxicity in mice right frontal lobes of *A. nancymae*, and stimulating mice producing IL-12 which activates *A. nancymae* lymphocytes in vitro [133]. This evidence supports M002 to be trailed in a phase 1 study for patients with rGBM.

#### 2.4.2. Adenovirus-Based

Adenovirus, a double-stranded, envelope-less DNA virus, is a common human pathogen that typically causes mild upper respiratory tract infections [146].

##### Adenoviral Vectors for Gene Therapy (HSV-tk)

The first human study using HSV-tk to treat rGBM was conducted in 1998 [167]. Twelve rGBM patients were injected with Herpes simplex virus type 1 thymidine kinase (HSV-1TK) mediated by retrovirus; after a 7-day transduction period, ganciclovir (GCV) was administered for 14 days. This treatment did not decline the quality of life. The median survival time was 206 days; one-fourth of them lived longer than one year. While tumor progression occurred in eight of them after four months from the treatment, the remaining four had significantly longer survival times. Their median survival was 528 days, compared with 194 days (*p* = 0.03). Another group reported an international, multicenter, open-label, uncontrolled phase II study using HSV-1TK and ganciclovir combination therapy in patients with relapsed GBM in 1999 [136]. After administering a suspension of retroviral vector-producing cells in participants who did tumor resection, they were injected with ganciclovir in the following 14 to 27 days. Overall, 48 patients were treated following the trial proposal in 11 centers in Europe and Canada. It showed the overall median survival time was 8.6 months, 13 of 48 (27%) survived over one year, seven patients had at least six months recurrence-free period, two patients with 12 months of progression-absence, and one remained recurrence-free at more than two years. There was no evidence of replication competent retrovirus in either peripheral blood leukocytes or tissue. One more similar trial was implemented in 2003 wherein Peter Sillevis Smitt et al. reported the safety of administration of 4.6 × 10^11^ adenoviral particles by 50 injections into the wound bed following the resection of recurrent gliomas [136].

Furthermore, a randomized control trial proved the efficacy of HSV-tk adenovirus (AdvHSV-tk) and GCV [155]. AdvHSV-tk was generated in a HEK293 cell line that stably expresses the E1 protein (ECACC, European Collection of Cell Cultures, Salisbury, UK). The study population consisted of 36 primary or recurrent GBM patients. The exposure group was assigned randomly and received AdvHSV-tk gene therapy (3 × 10^10^ pfu) after tumor resection; then, the intravenous ganciclovir was infused in 5 mg/kg twice daily for 14 days. In comparison, 19 patients in the control group followed standard care after radiotherapy. Finally, the median survival in the AdvHSV-tk group was 65% longer than the control group (62.4 vs. 37.7 weeks) and significantly longer than those in a historical control group (62.4 vs. 30.9 weeks). Moreover, the results showed no evidence of more prolonged survival requiring increasing concomitant medication use. Moreover, the treatment was well tolerated. In conclusion, AdvHSV-tk gene therapy and GCV are potentially efficient and safe treatments for primary or recurrent GBM.

##### Delta-24-RGD (DNX-2401)

Delta-24-RGD adenovirus (DNX-2401) is modified from Human adenovirus 5 (HAd5), which is deleted 24 base pairs in the E1A gene, and RGD-motif is inserted into the H-loop region of the adenovirus, thus enhancing the selective replication of the virus [139,156].

DNX-2401 conducted in rGBM treatment has only happened in the last few years. In 2018, the first report of a phase 1 study was published [139]. Thirty-seven patients were assigned to A (*n* = 25) and B groups (*n* = 12). On day 0, both groups executed stereotactic tumor injection of DNX-2401 (1 × 10^7^ to 3 × 10^10^ vp). Group A then followed up and assessed the toxicity and response, while group B did *en bloc* tumor resection along with catheter and intramural injection of DNX-2401 (1 × 10^7^ to 3 × 10^8^ vp) at day 14, alongside biological and toxicity studies. In group A, 72% (*n* = 18) of the patients showed tumor reductions with 9.5 months median survival duration. Moreover, five people survived longer than three years from the surgery, and three of five demonstrated a dramatic reduction (≥95%) in tumor size. Because of resection on day 14, group B can only provide survival information, wherein two of the twelve had more than two years of survival, and the OS was 13 months. Furthermore, DNX-2401 replicates and spreads within the tumor in group B, wherein a histopathologic check showed that CD8^+^ and T-bet^+^ cells infiltrated the tumor, indicating direct virus-induced oncolysis. It proved that DNC-2401 therapy caused direct oncolytic effects and anti-glioma response, which led to immune responses and long-term survival in patients with rGBM. In addition, a team of Japanese researchers found that patient-derived bone marrow human mesenchymal stem cells (PD-BM-hMSCs) loaded with Delta-24-RGD (PD-BM-MSC-D24) were able to both eradicate glioma tissues in vitro and improve the survival rate of mice harboring U87MG gliomas in vivo [140]. It provides evidence for using PD-BM-hMSCs to deliver DNX-2401 to treat brain tumors.

#### 2.4.3. Reovirus-Based

Reovirus, also named respiratory enteric orphan virus, is a naturally occurring double-stranded RNA virus. A phase 1 study indicated that after injecting reovirus at 1 × 10^7^, 1 × 10^8^, or 1 × 10^9^ tissue culture infectious dose 50 in a volume of 0.9 mL [141], Karnofsky’s Performance scores of seven patients increased without showing grade III or IV adverse events (AEs). Ten patients had tumor progression; the other two either remained stable or were not evaluable. The OS was 21 weeks (range: 6–234 weeks), with one of them alive at the discontinued point. Generally, a maximum dose was not reached, and the results demonstrated good tolerance to using these doses and schedules in patients with rGBM. A fellow dose escalation study was conducted in 2014 [142]. Fifteen adult patients were injected with 1 × 10^8^ to 1 × 10^9^ tissue culture infectious dose 50; two patients had stable disease as their best performance at the follow-up endpoint, one patient had a partial response, and twelve patients had tumor progression. For survival issues, thirteen patients survived approximately two years, and the remining two were alive in the following 3 and 5 years, respectively. 

#### 2.4.4. Newcastle Disease Virus Based

##### NDV-HUJ Oncolytic Virus

NDV is a single-stranded RNA virus whose natural host is poultry, and NDV-HUJ is the oncolytic HUJ strain of the Newcastle disease virus [143]. A phrase 1/2 study determined NDV-HUJ safety and tumor response. Initially, 14 patients were enrolled and completed an accelerated intrapatient dose-escalation protocol, from 0.1 to 11 billion infectious units (BIU) of NDV-HUJ (1 BIU = 1 × 10^9^ EID50 50% egg infectious dose). They then received the highest preclinical tested dosage (55 BIU) for three cycles. Secondly, the patients received two to three cycles of 11 BIU depending on their tumor progression. Grade I/II constitutional fever was the most common adverse effect, possibly related to treatment, among the patients. The maximum tolerated dose was not observed. These findings encouraged the continued evaluation of NDV-HUJ in rGBM.

#### 2.4.5. The Future Directions of Oncolytic Viral Therapy in (rGBM)

There are still many issues to be explored in treating rGBM with oncolytic viruses, including mechanism of action, safety and maximum dose, and mode of inoculation. The use of oncolytic viruses in combination with standard conventional therapeutic regimens and other agents, such as immune checkpoint inhibitors, will also be the focus of further research. In addition, OVs can also serve as innate adjuvants to enhance antitumor immune response and combine with other immunotherapies to improve the immunosuppressive microenvironment. In the future, OVs and related combination therapeutic strategies to improve the outcome of glioma treatment are promising.

### 2.5. Combination Strategies for GBM

#### 2.5.1. Chemotherapy and Radiotherapy

In 1970, there was clinical evidence that patients with GBM with lomustine plus radiotherapy achieved median survival of 11.5 months, which was longer than that of patients receiving radiotherapy alone [157]. Subsequently, it was found that TMZ was treated concurrently with radiotherapy, and maintenance chemotherapy for 6 weeks improved the survival of GBM patients to 14.6 months [158]. A large number of clinical trials have been conducted in people under 60 to 70 years of age, so most clinicians consider TMZ plus radiotherapy to be the standard of care for GBM patients under 65 years of age. In recent years, there have been many experimental data from elderly patients that have also demonstrated better results during TMZ added to radiotherapy [4,159,160]. At the same time, there are results that support TMZ therapy for longer survival in patients with MGMT promoter methylation tumors [161]. This suggests that the status of MGMT can be used to select patients who benefit more from treatment, avoiding toxic and expensive treatment for patients with poor prognosis. Especially in older patients, individualized treatment should be based on performance status, degree of resection of the lesion, and MGMT status, including radiation dose and whether or not to combine chemotherapy [159,163]. However, TMZ treatment has limitations. Combination chemotherapy and radiation therapy can lead to comorbidities, including bone marrow suppression and infection. Common side effects are neutropenia and thrombocytopenia [164]. There is no evidence that changing the dose of the TMZ regimen or extending its administration beyond 6 months improves survival. Furthermore, the effect of TMZ is correlated with MGMT promoter methylation. Chemical resistance to alkylating agents in GBM patients leads to research to explore more targeted treatments, such as exploring new drugs including O6-benzylguanine (O6-BG) and O6-(4-bromothenyl) guanine (O6-BTG), RNAi, and viral proteins targeting MGMT to improve the anti-tumor effects of TMZ [165].

#### 2.5.2. Molecularly Targeted Drugs

Bevacizumab (BVZ) was approved in 2009 in countries such as the United States and Switzerland for the treatment of rGBM, but data from two large phase III European Organization for Research and Treatment of Cancer (EORTC) trials did not show that it extended OS in patients with GBM [166]. However, it has significantly improved PFS rates and reduced demand for steroids, which can improve quality of life [168]. Much of the current research is looking for a combination of BVZ and immunomodulators or other drugs.

#### 2.5.3. Tumor Treatment Fields (TTFields)

The Phase III registration trial demonstrated that TTFields has the same efficacy as chemotherapy and bevacizumab, and the Food and Drug Administration (FDA) approved the TTFields for the treatment of rGBM [169]. Since then, multiple clinical trials have shown that TTFields have better results in combination with surgery and chemoradiotherapy. The trial of Felix Bokstein et al. confirmed that TTFields in combination with chemotherapy and radiation therapy has a good effect and does not increase the toxicity of chemotherapy or radiotherapy. In addition to the appearance of adverse effects of scalp irritation, this combination therapy is safe and feasible. They are preparing to conduct a phase II study to further test the protocol [170]. Experiments on newly diagnosed GBM patients have shown that TTFields combined with TMZ and CCNU is safe and feasible, and has potentially beneficial therapeutic effects [171]. Clark et al. found through in vitro cell experiments that the antitumor efficacy of TTFields was not affected by the MGMT status of cells [172]. The most common adverse effect of this therapy is localized skin disease, but it causes much less hematological toxicity and gastrointestinal irritation than radiotherapy and chemotherapy. The use of dexamethasone may reduce the therapeutic effect of TFields and radiotherapy. Gregory’s research illustrates that placement of TTFields arrays does not significantly affect target volume coverage [174]. The modeling results of Eric et al. show that the therapeutic effect of TTFields is limited by the location of the tumor in the brain, and larger tumors may require longer treatment times [169].

Although the Non-immunotherapy combination therapy is not the point that we discuss in this review, as an important part of therapy for GBM, the Non-immunotherapy combination therapy still could be summarized in the Table 5.

#### 2.5.4. Combination Strategies of Immunotherapy

Immunotherapy has made advances in the treatment of rGBM patients, however, there are several causes that make single immunotherapy treatments less successful.

Due to the intricately regulated immune system in rGBM, inhibiting the PD-1/PD-L1 pathway alone in rGBM is insufficient to activate sufficient effector T cells to destroy tumor cells in rGBM [175,176,177,178,179,180,181]. Additionally, adaptive resistance to anti-PD-1 therapy, such as the exhaustion of cytotoxic T cells brought on by coinhibitory molecules, results in unfavorable therapies [97,182]. In addition, despite the fact that anti-PD-1 therapy can kill certain tumor cells, many subclonal tumor cells are able to survive and grow continuously as a result of the complex and varied biological characteristics of rGBM [101,175].

CAR-T cell therapy has been developed as an effector for lymphocytes to increase immune response in GB because the blood-brain barrier makes it challenging for immune cells and medications to enter tumor tissues in rGBM [102]. CAR-TR cells, on the other hand, have limited infiltration and a brief lifetime, which results in a low cytotoxic impact on curing rGBM [40,183,184]. The heterogeneity of tumor cells is also blamed for contributing to the recurrence of rGBM [101,175].

Although numerous tumor vaccines have been proposed to treat rGBM, there are a number of obstacles that hinder vaccinations from working [185,186,187,188,189]. For instance, GB is characterized as lacking efficient treatment targets due to its poor immunogenicity and tumor [190] mutational burden [191]. High rGBM heterogeneity and difficult activated cytotoxic cell transition through blood-brain barriers are also attributed to vaccine treatment failure [101,175].

As a result, combined immunotherapy is used as a treatment option for rGBM more successfully. In comparison to DC vaccination alone, it has been found that anti-PD-1/PD-L1 medication dramatically enhances the immunological response of rGBM patients following vaccination. The mechanism may be that the DC vaccine increases PD-1 expression, and that anti-PD-1 therapy administered after the DC vaccine increases its efficacy and promotes tumor cell cytolysis [192]. Furthermore, EGFRvIII-specific CAR-T cell therapy was found to be beneficial in the treatment of rGBM patients when combined with anti-PD-1/PD-L1 therapy [56]. Also, there is the experimental proof that using anti-PD-1 and CD19 CAR-T cells together dramatically increased therapeutic success in refractory diffuse large B-cell lymphoma [193]. Based on these findings, when paired with anti-PD-1/PD-L1 therapy, CAR-T cell therapy targeting additional peptides, such as IL-13R2, EphA2, or HER2, may be beneficial for treating rGBM as well.

In addition to PD-1, other marker genes downregulating T cell function on the surface of T lymphocytes may serve as inhibitory receptors, including CTLA-4, TIM-3, and LAG-3 [194,195,196,197]. LAG-3, a T cell exhaustion marker that is abundantly expressed in GB tumors, is one example. Anti-LAG-3 antibodies may thus be used in combination with other ICIs to treat rGBM. A study has discovered that compared with the control group in mice, the mice in the anti-LAG-3 combined with anti-PD-1 therapy achieved a significant improvement regarding survival benefits [198]. Additionally, several clinical studies and experiments are being conducted to investigate the effectiveness of combination treatments that simultaneously target CTLA-4, LAG-3, TIM-3, and PD-1/PD-L1 [199,200]. Costimulatory molecules are highly expressed in T lymphocytes, including 4-1BB and OX40, and they can also be utilized to combine anti-PD-1 antibodies to effectively treat rGBM [201,202,203,204,205].

There are also several studies investigating combination therapy to increase the positive effects of vaccinations, taking into account how chemotherapy and DC vaccines might complement one another. For instance, it has been shown that chemotherapy given after vaccination considerably increased the survival duration of rGBM when compared to chemotherapy or vaccine given alone [206].

Depending on whether the rGBM is positioned in a resectable anatomical site, surgical resection and reradiation are also effective treatment options for the condition [207]. More importantly, it has been discovered that radiation and surgical resection dramatically improve rGBM when used in conjunction with other treatments like anti-PD-1 therapy. There is evidence that treating rGBM with neoadjuvant anti-PD-1 therapy plus surgical resection, and subsequently, adjuvant anti-PD-1 therapy, is an effective strategy. Neoadjuvant anti-PD-1 therapy helps to stop the progression of the cell cycle and proliferation by triggering the IFN-γ response. Resection was performed to reduce the tumor burden and maintain tumor-specific T cell function. Adjuvant anti-PD-1 therapy helps to further kill any remaining tumor cells in the rGBM [208]. The combination of radiotherapy and anti-PD-1 therapy, known as neoadjuvant anti-PD-1 plus radiotherapy, followed by adjuvant anti-PD-1 therapy, has been shown to have a synergistic effect on tumors. This is due to how radiotherapy can accelerate the clinical effect of anti-PD-1 on tumors via activating immunogenic cell death and TCR diversity with increased IFN-γ release [209,210].

#### 2.5.5. Virus-Based Combination Strategies

Though each treatment received positive responses in some patients, some still suffered AEs or died, attributed to tumor progression. Thus, in 2003, Isabelle M. Germano et al. combined Adenovirus, Herpes simplex-thymidine kinase, and ganciclovir [144]. At the time of recurrence, researchers performed tumor resection and injected ADV/HSV-tk complex in the tumor bed, then administered GCV (10 mg/kg/day) within 24 h after surgery for seven days. Eleven patients were assigned to three sub-cohorts, who received 2.5 × 10^11^, 3.0 × 10^11^, and 9.0 × 10^11^ VP ADV/HSV-tk complex, respectively. Three months later, 8/10 patients’ Karnofsky score was maintained ≥70 and 5/9 in 6 months. Ten of eleven patients survived longer than 52 weeks, the average survival was 112.3 weeks, and one patient was still alive 248 weeks after diagnosis. This indicated that the used doses of the complex were safe and that the whole treatment schedule was tolerable. 

#### 2.5.6. The Current Situation and the Prospect of Combination Strategies for GBM

Currently, the FDA has approved five drugs and one device to treat GBM: TMZ, lomustine, intravenous carmustine, carmustine wafer implants, BVZ, and TTFields. The radiotherapy and TMZ chemotherapy are considered to be the standard of care for GBM. TTFields is the only treatment which has been shown improved OS (20.5 vs. 15.6 months) compared to the current standard of care [211], but has not been universally accepted as a part of standard of care. Bevacizumab is the only FDA approved drug for recurrent GBM. More research should be conducted to find the real SOC for recurrent GBM.

There are still numerous issues to be resolved even though combination immunotherapy for rGBM has shown promising outcomes. To maximize treatment effectiveness, the best combination immunotherapy sequence should first be confirmed. For instance, it was discovered that administering an anti-PD-1 antibody after an agonist anti-OX40 antibody could increase its effectiveness in preventing tumor growth, but administering both antibodies at the same time could counteract the antitumor effects of an agonist anti-OX40 alone in the rGBM model [204,205]. Second, the timing of immunotherapy is crucial and should be confirmed when used in conjunction with other forms of treatment. Take LAG-3 as an example, combined anti-LAG-3 on 10th day with anti-PD-1 therapy achieved an unideal survival benefit compared to the mice in the combined anti-LAG-3 on the 7th day with anti-PD-1 therapy, suggesting anti-LAG-3 are more effective in the early stage of the tumor when combined with other immunotherapy [198]. Third, compared to immunotherapy alone, there are many more combination tactics available, and validating each potential immunotherapy combination approach requires an inordinate amount of time and money. Therefore, massive parallel combination immunotherapy arrays or computational immunotherapy combination methods are needed urgently in the community to decrease costs significantly in discovering promising combination immunotherapy. Fourth, finding efficient therapeutic biomarkers is necessary to direct the development of efficient combination immunotherapy. Due to the great heterogeneity and low immunogenicity of rGBM, it will be advantageous to find reliable and attractive molecular targets for possible combination immunotherapy. With the development of technology, there are more and more good practices to identify and evaluate potential disease-associated therapeutic molecular targets and develop prediction methods to predict the efficacy of combination therapy in common diseases [212,213,214,215,216]. Last but not least, developing strategies to maximize CAR-T cell longevity, increase cell infiltration, and circumvent blood-brain barrier issues are effective directions to increase the survival of rGBM patients after receiving combination immunotherapy.

## 3. Conclusions and Future Perspective of Immunotherapy to Recurrent GBM

Immunotherapy has emerged as a promising strategy for the treatment of GBM, as it seeks to harness the power of the immune system to recognize and attack cancer cells. Several types of immunotherapies have been studied for GBM, including CAR-T, checkpoint inhibitors, cancer vaccines, oncolytic viruses, and combination strategies. (Figure 1).

Checkpoint inhibitors target proteins that regulate the immune system, such as PD-1 and CTLA-4, and can enhance the ability of T cells to attack cancer cells. Cancer vaccines can prime the immune system to recognize and attack cancer cells, and can even be developed using a patient’s own tumor cells to generate a personalized vaccine. CAR-T therapy can specifically target cancer cells by isolating and multiplying T cells, while oncolytic viruses infect and destroy cancer cells.

To improve the efficacy of immunotherapy for GBM, several approaches are being explored. As the future direction of immunotherapy for recurrent GBM, combination therapies will likely involve a combination of different approaches, which aim to target multiple pathways involved in cancer growth and immune evasion, as these have shown promise. Despite the promise of immunotherapy for GBM, clinical trials have had mixed results. Some studies have shown modest improvements in survival and quality of life, while others have not shown significant benefits over traditional therapies. The heterogeneity of GBM, as well as the immunosuppressive tumor microenvironment, may play a role in the variable response to immunotherapy.

While there is still much to learn about the optimal use of immunotherapy for recurrent GBM, the field holds great promise for improving outcomes and quality of life for patients with this devastating disease. Continued research is needed to address the challenges and identify the most effective combination of immunotherapy approaches, as well as to develop new biomarkers and delivery methods to improve outcomes for patients with recurrent GBM.

## Figures and Tables

**Figure 1 cancers-15-04308-f001:**
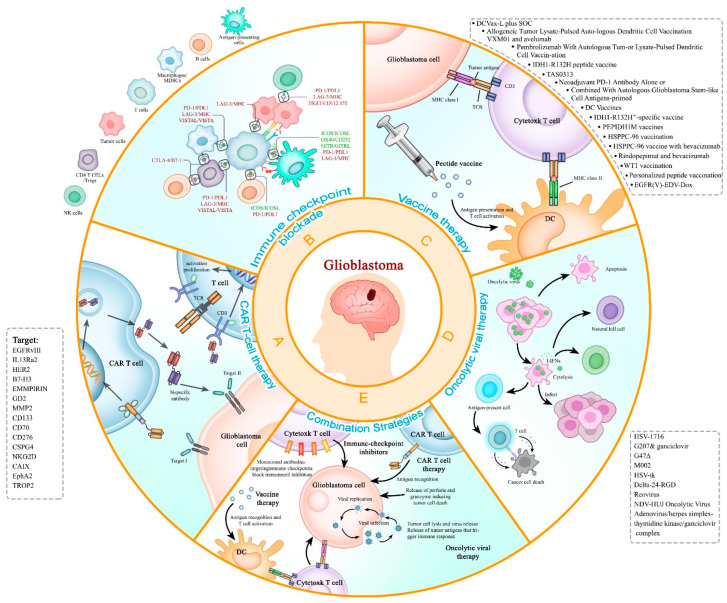
The overview of immunotherapy for the treatment of recurrent glioblastoma. A: CAR-T therapy can target antioens that are highly expressed on the GBM cell suraces, including EGFRvIII, IL13Ra2, HER2, B7-H3, EMMPIRIN, GD2, MMP2, CD133, CD70, CD276, CSPG4, NKG2D, CAIX, EphA2, TROP2. B: lmmune checkpoint inhibitors are monoclonal antibodies that target immune checkpoints to block immune cell inhibition, such as IDO1, CTLA-4/B7, LAG-3, CD137, VISTA, Siglec-15, HHLA2, and LAG-3/MHC, TIM3/CEACAM-1, VISTAL/VISTA, TIGIT/CD155, CD112 for negative immune regulation, ICOS/ICOSL, OX40/CD252, GITR/GITRL, CTLA-4/CD80 or CD86 for positive immune regulation. C: Vaccine therapy depends on dendritic cells, which present antigens or peptides to cytotoxic T cells via MHC class I-TCR interaction leading to T cell activation. The cytotoxic T cells then eradicate GBM cells via MHC class -TCR interaction, with vaccine therapy for recurrent GBM including DCVax-L plus SOC, Allogeneic Tumor Lysate-Pulsed Autologous Dendritic Cell Vaccination, VXM01 (DNA plasmid vaccine for VEGFR-2) and avelumab (anti-PD-L1), Pembrolizumab With Autologous Tumor Lysate-Pulsed Dendritic Cell Vaccination, IDH1-R132H peptide vaccine, TAS0313, Neoadjuvant PD-1 Antibody Alone or Combined With Autologous Glioblastoma Stem-like Cell Antigens-primed DC Vaccines, IDH1-R132H^+^-specific vaccine, PEPIDH1M vaccines, HSPPC-96 vaccination, HSPPC-96 vaccine with bevacizumab, Rindopepimut and bevacizumab, WT1 vaccination, Personalized peptide vaccination, EGFR(V)-EDV-Dox. D: Oncolytic viral therapy utilizes genetically engineered viruses, which could selectively infect and replicate in GBM cells, resulting in cell lysis and release of tumor antigens. This can further trigger an adaptive antitumor immune response by stimulating antigen presenting cells, which include HSV-1716, G207, ganciclovir, G47Δ, M002, HSV-tk, Delta-24-RGD, Reovirus, NDV-HUJ Oncolytic Virus, Adenovirus/herpes simplex-thymidine kinase/ganciclovir complex. E: As the future direction of immunotherapy for recurrent GBM, combination strategies could be the future direction of immunotherapy for recurrent GBM, which involve a combination of different therapies, have shown more promising outcomes than single therapy.

**Table 1 cancers-15-04308-t001:** The clinical trials of CAR-T therapy on rGBM in ClinicalTrials.gov.

Project Name	Target	Clinic Phase	Start Date	Estimated or Actual Completion Date	Estimated or Actual Enrollment	Status
NCT00730613	IL-13Rα2	Phase 1	2 Feb. 2002	11 Aug. 2011	3 participants	done
NCT01082926	IL-13Rα2	Phase 1	10 May 2010	1 Sep. 2013	6 participants	done
NCT02208362	IL-13Rα2	Phase 1	15 May 2015	18 Jun. 2023	82 participants	going
NCT04003649	IL-13Rα2	Phase 1	19 Dec. 2019	31 Dec. 2023	60 participants	going
NCT02209376	EGFRvIII	Phase 1	14 Nov. 2014	1 Apr. 2018	11 participants	done
NCT01454596	EGFRvIII	Phase 1	12 May 2012	1 May 2012	18 participants	done
NCT03726515	EGFRvIII	Phase 1	11 Mar. 2019	27 Feb. 2021	7 participants	done
NCT05024175	EGFRvIII and EGFR	Phase 1	1 Dec. 2021	1 Aug. 2039	18 participants	going
NCT05168423	EGFR and IL13Rα2	Phase 1	19 Mar. 2023	19 Dec. 2029	18 participants	going
NCT01109095	HER2	Phase 1	1 Oct. 2010	1 Mar. 2018	16 participants	done
NCT03389230	HER2	Phase 1	14 Aug. 2018	15 Dec. 2023	42 participants	going
NCT03383978	HER2	Phase 1	1 Dec. 2017	31 Dec. 2023	42 participants	going
NCT04045847	CD147	Phase 1	30 Oct. 2020	30 May 2022	31 participants	Unknown
NCT05627323	MMP2	Phase 1	1 Feb. 2023	1 Jan. 2041	42 participants	going
NCT04214392	MMP2	Phase 1	26 Feb. 2020	26 Feb. 2020	36 participants	going
NCT04385173	B7-H3	Phase 1	1 Dec. 2022	1 May 2024	12 participants	going
NCT05241392	B7-H3	Phase 1	27 Jan. 2022	31 Dec. 2024	30 participants	going
NCT04077866	B7-H3	Phase 1/2	1 Jun. 2023	1 Aug. 2025	40 participants	going
NCT05366179	B7-H3	Phase 1	2 Sep. 2022	May 2030	36 participants	going
NCT05474378	B7-H3	Phase 1	12 Jul. 2022	1 Aug. 2025	39 participants	going
NCT05353530	CD70	Phase 1	1 Oct. 2022	Dec. 2040	18 participants	going
NCT04717999	NKG2D	Unknown	1 Sep. 2021	21 Dec. 2023	20 participants	going

**Table 2 cancers-15-04308-t002:** Ongoing rGBM trails combined with PD-1/PD-L1 blockades.

Clinical Trail	Phase	Interventions	Arms	Combined Therapy
NCT05700955	I	Drug: pembrolizumab and TMZ	Single arm: neoadjuvant pembrolizumab + TMZ	Neoadjuvant chemotherapy
NCT03661723	II	Drug: pembrolizumab, bevacizumab Radiation: re-RT	Arm 1: pembrolizumab + RT (lead-in) Arm 2: pembrolizumab + bevacizumab + RT (lead-in) Arm 3: pembrolizumab + RT Arm 4: pembrolizumab + bevacizumab + RT	Adjusted RT, VEGFA inhibitor
NCT03743662	II	Drug: pembrolizumab, bevacizumab Radiation: re-RT Procedure: re-resection	Arm 1: re-RT + bevacizumab + Nivolumab Arm 2: re-RT + bevacizumab + Nivolumab + re-resection	re-RT, bevacizumab, re-resection
NCT04977375	I/II	Drug: pembrolizumab radiation: stereotactic RT	Single arm: pembrolizumab + stereotactic RT + surgical resection	Stereotactic RT
NCT02866747	I/II	Drug: durvalumab Radiation: HFSRT	Arm 1: RT alone Arm 2: RT + durvalumab	HFSRT
NCT02829931	I	Radiation: HFSRT Drug: nivolumab, bevacizumab, ipilimumab	Single arm: HFSRT + ipilimumab + nivolumab + bevacizumab	VEGFA, CTLA-4 inhibitors, HFSRT
NCT03722342	I	Drug: TTAC-0001, pembrolizumab	Arm 1: TTAC-0001 12 mg/kg on D1, D8 and D15 + pembrolizumab 200 mg on D1 Arm 2: TTAC-0001 16 mg/kg on D1, D8 and D15 + pembrolizumab 200 mg on D1 Arm 3: TTAC-0001 8 mg/kg on D1, D8 and D15 + pembrolizumab 200 mg on D1	VEGFR2 inhibitor
NCT02311582	I/II	Drug: pembrolizumab Procedure: LITT	Arm 1: pembrolizumab + LITT Arm 2: pembrolizumab only	Thermotherapy
NCT03277638	I/II	Drug: pembrolizumab Procedure: LITT	Single arm: pembrolizumab + LITT	Thermotherapy
NCT03341806	I	Drug: avelumab Procedure: LITT	Arm 1: avelumab Arm 2: avelumab + LITT	Thermotherapy
NCT03430791	I/II	Drug: nivolumab, ipilimumab Device: TTF	Arm 1: nivolumab + TTF Arm 2: nivolumab + ipilimumab +TTF	CTLA-4 inhibitor, tumor treating fields
NCT03532295	II	Drug: epacadostat, retifanlimab, bevacizumab Radiation: RT	Arm 1: retifanlimab + RT + bevacizumab Arm 2: retifanlimab + RT + bevacizumab + epacadostat	RT, VEGFA, and IDO1 inhibitor
NCT02794883	II	Drug: durvalumab, tremelimumab	Arm 1: durvalumab Arm 2: durvalumab + tremelimumab Arm 3: tremelimumab	CTLA-4 inhibitor
NCT03493932	I	Drug: BMS-986016, nivolumab	Single arm: BMS-986016 + nivolumab	LAG-3 inhibitor
NCT02658981	I	Drug: BMS-986016, urelumab, nivolumab	Arm 1: BMS-986016 Arm 2: BMS-986016 + nivolumab Arm 3: urelumab + nivolumab	LAG-3, CD137 inhibitors
NCT05465954	II	Drug: efineptakin alfa, pembrolizumab	Single arm: efineptakin alfa + pembrolizumab, before and after surgery	Neoadjuvant IL7
NCT04201873	I	Biological: DC tumor cell lysate vaccine Drug: pembrolizumab, poly ICLC	Arm 1: pembrolizumab + ATL-DC + poly ICLC Arm 2: placebo + ATL-DC + poly ICLC	DC vaccine
NCT04013672	II	Drug: pembrolizumab, surVaxM, sargramostim, montanide ISA 51	Arm 1: have not received immunotherapy Arm 2: have failed prior anti-PD1 therapy	Peptide-based vaccine
NCT03665545	I/II	Drug: IMA950/Poly-ICLC and pembrolizumab	Arm 1: IMA950/Poly-ICLC Arm 2: IMA950/Poly-ICLC + pembrolizumab	Peptide-based vaccine
NCT05084430	I/II	Drug: M032, pembrolizumab	Single arm: pembrolizumab + M032	Oncolytic herpes simplex virus
NCT04479241	II	Drug: lerapolturev, pembrolizumab	Single arm: lerapolturev + pembrolizumab	Oncolytic poliovirus
NCT02798406	II	Biological: DNX-2401 Drug: pembrolizumab	Single arm: DNX-2401 + pembrolizumab	Oncolytic adenovirus
NCT05463848	II	Drug: pembrolizumab, olaparib, TMZ	Arm 1: pembrolizumab + olaparib + TMZ Arm 2: pembrolizumab monotherapy	PARP inhibitor, chemotherapy
NCT02430363	I/II	Drug: pembrolizumab Biological: suppressor of the PI3K/Akt pathways	Single arm: pembrolizumab + suppressors of the PI3K/Akt pathways	PI3K/Akt suppressors
NCT05053880	I/II	Drug: ACT001, pembrolizumab	Arm 1: pembrolizumab Arm 2: pembrolizumab+ACT001	PAI-1 inhibitor

TMZ: temozolomide, RT: radiation therapy, HFSRT: hypofractionated stereotactic irradiation, LITT: laser interstitial thermotherapy, TTF: Tumor Treating Fields, DC: dendritic cell.

**Table 3 cancers-15-04308-t003:** The latest clinical trials on vaccination therapies for rGBM.

Type	Last Reported	Therapy	Phase	Registration Number
DC vaccines	2023	Allogeneic Tumor Lysate-Pulsed Autologous Dendritic Cell Vaccination	Early Phase I	NCT03360708
Peptide vaccines	2023	Allogeneic tumor lysate vaccine	Phase I	NCT04642937
Nucleic acid vaccines	2022	VXM01 (DNA plasmid vaccine for VEGFR-2) and avelumab (anti-PD-L1)	Phase I/II	NCT03750071
DC vaccines	2022	DCVax-L plus SOC	Phase III	NCT00045968
DC vaccines	2022	Pembrolizumab With Autologous Tumor Lysate-Pulsed Dendritic Cell Vaccination	Phase I	NCT04201873
DC vaccines	2022	mRNA tumor antigen-pulsed autologous DCs	Phase I	NCT02808364
Peptide vaccines	2022	TAS0313	Phase II	JapicCTI-183824
Peptide vaccines	2022	VBI-1901 (targeting CMV antigen gB and pp65)	Phase I/II	NCT03382977
DC vaccines	2021	Neoadjuvant PD-1 Antibody Alone or Combined with Autologous Glioblastoma Stem-like Cell Antigens-primed DC Vaccines	Phase II	NCT04888611
DC vaccines	2021	allogeneic glioblastoma stem-like cell line-pulsed DC cell	Phase I	NCT02010606
Peptide vaccines	2021	PEPIDH1M vaccines	Phase I	NCT02193347
Peptide vaccines	2021	HSPPC-96 vaccine	Phase II	NCT00293423
Peptide vaccines	2021	HSPPC-96 vaccine with bevacizumab	Phase II	NCT01814813
Peptide vaccines	2020	Rindopepimut and bevacizumab	Phase II	NCT01498328
Peptide vaccines	2020	HSPPC-96 vaccine	Phase I	NCT02722512
DC vaccines	2020	Autologous tumor cell-pulsed DCs (ADCTA)	Phase III	NCT04277221
Peptide vaccines	2019	Personalized peptide vaccination	Phase III	AMED number: 16ck0106086h0003
Nucleic acid vaccines	2019	EGFR(V)-EDV-Dox	Phase I	NCT02766699
DC vaccines	2019	Autologous tumor lysate-loaded DCs	Phase I	NCT04002804
DC vaccines	2019	Tumor lysate-pulsed DCs	Phase II	NCT00576537
DC vaccines	2019	GSC (Glioma Stem Cells) -Loaded Dendritic Cells	Phase I	NCT02820584

**Table 4 cancers-15-04308-t004:** Oncolytic viral therapy trial in recurrent GBM (rGBM).

Agents	Year	Study Design	Subjects	Experiment Time	Registration Number
Herpes simplex virus (HSV-1716)	2000	Phrase I trial	Patients had biopsy proven high grade glioma	24 months	PMID10845724 [148]
G207	2009	Phrase I b trial	Patients had an initial histologically confirmed diagnosis of glioblastoma multiforme	19 months	F05041106 [128]
G207	2014	Phrase I trial	Patients had pathologically confirmed residual/recurrent glioblastoma multiforme, gliosarcoma, or astrocytoma	11–51 months	NCT00157703 [129]
G207	2015	Case report	A 52-year-old Caucasian female had a GBM with an infltrative glial tumor	More than 5.5 years	NCT00028158 [130]
G207	2022	Cross-sectional study (a Gene Expression Analyses)	Patients are from the phase Ib G207 clinical trial (NCT00028158)	/	/[131]
G47Δ	2022	Phrase II trial	Patients who had a pathologically confirmed diagnosis of glioblastoma with a persistent or recurrent tumor	2–5 years	UMIN000015995 [132]
Herpes simplex virus Expressing Interleukin-12 (M002)	2012	Animal experiment	Specific-pathogen-free female SCID and B6D2F1 mice	More than 80 days	/[133]
Herpes simplex virus type 1 thymidine kinase suicide gene therapy (HSV1-tk)	1998	Phrase I/II trial	Patients had a recurrence of primary glioblastoma	830 days	/[135]
Herpes simplex thymidine kinase gene (HSV-tk)	1999	Phrase II trial	Patients with relapsed GBM	More than 15 months	/[136]
Adenovirus mediated HSV-tk gene therapy (AdvHSV-tk)	2004	RCT	All patients with operable primary or recurrent highgrade glioma	More than 200 weeks	/[138]
Delta-24-RGD	2018	Phrase I trial	Patients with recurrent malignant glioma	More than 3 years	NCT00805376 [139]
Delta-24-RGD	2022	Animal experiment	95 mice	More than 100 days	/[140]
Reovirus	2008	Phrase I trial	Patients had a diagnosis of GBM	More than 234 weeks	/[141]
Reovirus	2014	Phrase I trial	Patients had either first, second, or third occurrence of a supratentorial tumor with a histologic diagnosis consistent with glioblastoma multiforme	More than 989 days	/[142]
NDV-HUJ Oncolytic Virus	2005	Phase I/II Trial	Patients had been diagnosed with GBM based on histology and gadolinium-enhanced (Gd+) MRI, and all had a recurrence of GBM	More than 66 weeks	/[143]
G207& ganciclovir	2000	Animal experiment	Six-week-old female A/J mice	More than 30 days	/[145]
Adenovirus/herpes simplex-thymidine kinase/ganciclovir complex	2003	Phase I Trial	Patients had histologically confirmed malignant glioma, defined as GBM	More than 248 weeks	/[144]

**Table 5 cancers-15-04308-t005:** Non-immunotherapy combination therapy for GBM.

Clinical Trails	Phase	Interventions	Arms	Combined Therapy
NCT00684567	II	Drug: TMZ Radiation: RT	Single arm: TMZ + RT	Chemotherapy and radiotherapy
NCT01730950	II	Biological: BVZ Radiation: RT	Arm 1: BVZ Arm 2: BVZ + RT	Radiation therapy with bevacizumab for the rGBM
NCT01894061	II	Biological: BVZ Device: NovoTTF-l00A Other: Quality of Life Assessment	Arm 1: BVZ + NovoTTF-100A	NovoTTF-100A With Bevacizumab (Avastin) for the rGBM
NCT01849146	I	Drug: Adavosertib, TMZ Radiation: RT	Arm 1: Adavosertib + TMZ + RT Arm 2: adavosertib + TMZ	Adavosertib, RT, and TMZfor the Newly Diagnosed GBM or rGBM
NCT00650923	I	Drug: Ziv-aflibercept, TMZ, Procedure: RT, pharmacological study, laboratory biomarker analysis	Arm 1: ziv-aflibercept + RT + TMZ	Aflibercept, RT, and TMZ for the Newly Diagnosed GBM or rGBM

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
