# Peer review of "The Development of Immunotherapy for the Treatment of Recurrent Glioblastoma"

_cancers, 2023, doi:10.3390/cancers15174308_

Round 1

Reviewer 1 Report (Previous Reviewer 3)

The authors have addressed the comments and the manuscript could be considered for publication.

Quality of English language is okay. The sentence structure and syntax could certainly be improvised to convey the authors intended message.

Reviewer 2 Report (Previous Reviewer 2)

the authors adressed the reviewer´s comment adequately.

This manuscript is a resubmission of an earlier submission. The following is a list of the peer review reports and author responses from that submission.

Round 1

Reviewer 1 Report

In this review the authors try to summarize recent developments of immunotherapeutical approaches  for the treatment of recurrent glioblastoma. Although all different treatment modalities including CAR-T cell therapy, application of checkpoint inhibitors, vaccination strategies and oncolytic virus therapy are discussed, this review is very difficult to understand and almost not readable. Actually, it is a pure line-up of different preclinical and clinical studies with a lot of redundancy and also conflicting statements (more than 13000 words). This review should be completely restructured and would benefit from a more concise version with a clear summary at the end of each chapter.

Reviewer 2 Report

The authors present a review of immunotherapy in glioblastomas. In recent years, immunotherapy has emerged as a promising strategy for the treatment of recurrent glioblastoma by stimulate the body's immune system to recognize and attack cancer cells , which could be used as a in combination with other treatments such as surgery, radiation, and chemother- apy to improve outcomes for patients with recurrent glioblastoma, This therapy combines several key methods such as the use of monoclonal antibodies, chimeric antigen receptor T cell (CAR-T) therapy, checkpoint inhibitors, oncolytic viral therapy cancer vaccines, and combination strategies. In the authors´ review, they describe the latest immunotherapies for the treatment of glioblastoma and focus on the rGBM especially. They discuss the application state of latest immunotherapy for rGBM, provide the future direction and important challenges of each kind of immunotherapy methods.

The manuscript is well written. The authors researched the literature and topic well. The authors´present a hughe amount of literature and studies which were discussed and illustrated. 

IHowever, I suggest to discuss also pitfalls and negative results as well to explore all aspects of immunotherapy. Furthermore, I suggest to include more details or data of the daily clinical routine and practice of this therapy.

Reviewer 3 Report

The objective of the authors was to provide a comprehensive review on latest immunotherapies used for the treatment of glioblastoma with specific focus on recurrent glioblastoma (rGBM). The authors have discussed different immunotherapies including CAR-T, immune Checkpoint inhibitors, cancer vaccines (peptide, immune cell, and nucleic acid-based vaccines), oncolotypic viruses, and combination treatment strategies. The authors have also provided brief summary of clinical trials and animal studies using different therapies in rGBM. It is very interesting review article, but the manuscript could not be considered for publication in the current form for following reasons.

Concerns:

1) Please include the registration number for DC cell vaccines type and DCVax-L plus SOC therapy.

2) Include the registration number for peptide vaccines type and IDH1-R132H+-specific vaccine therapy in Table 3.

3) In addition, there are couple more therapies in peptide vaccines type, the registration numbers need to be updated. Please update the table 3 with relevance information.

4) Include experiment time for cross sectional study in 2022 in Table # 4.

5) Please replace "HER2 was found to overexpression in.." with ""HER2 was found to overexpress in.." or "Overexpression of HER2 was reported in ...". on page # 4.

6) Please include the reference for the statement "...focused on combining anti-PD-1/PD-L1...". and delete "Error! Reference source not found" text on page # 8. 

7) Please address incomplete sentences "The mianly vaccine for rGBM in the Error! Reference source not found..."

8) The font size is different in section 2.1.1 compared to text in the remaining sections of the manuscript. Could authors please fix the issue.

9) The alignment is not correct for text in the Last reported column in Table 3. Please fix the issue.

10) Please complete the sentence "The clinical trials and animal............ in Error! Reference source not found" in subsection 2.5 on page # 18.

11) Table 5 is mistakenly labeled as Table 3 on Page # 24. Please fix it.

12) Could authors please carefully proofread the manuscript and correct syntax errors.